# Optimized Two-Stage Anomaly Detection and Recovery in Smart Grid Data Using Enhanced DeBERTa-v3 Verification System

**DOI:** 10.3390/s25134208

**Published:** 2025-07-05

**Authors:** Xiao Liao, Wei Cui, Min Zhang, Aiwu Zhang, Pan Hu

**Affiliations:** 1State Grid Information and Telecommunication Group Co., Ltd., Beijing 100029, China; cuiwei@sgitg.sgcc.com.cn (W.C.); zhangmin@sgitg.sgcc.com.cn (M.Z.); zhangaiwu@sgitg.sgcc.com.cn (A.Z.); 2School of Automation Science and Engineering, Xi’an Jiaotong University, Xi’an 710049, China; 2206515221@stu.xjtu.edu.cn

**Keywords:** two-stage anomaly detection, smart grid security, DeBERTa-v3, transformer architecture, ensemble verification, time series analysis, cyber-attack detection, data recovery, TimER, generative models

## Abstract

The increasing sophistication of cyberattacks on smart grid infrastructure demands advanced anomaly detection and recovery systems that balance high recall rates with acceptable precision while providing reliable data restoration capabilities. This study presents an optimized two-stage anomaly detection and recovery system combining an enhanced TimerXL detector with a DeBERTa-v3-based verification and recovery mechanism. The first stage employs an optimized increment-based detection algorithm achieving 95.0% for recall and 54.8% for precision through multidimensional analysis. The second stage leverages a modified DeBERTa-v3 architecture with comprehensive 25-dimensional feature engineering per variable to verify potential anomalies, improving the precision to 95.1% while maintaining 84.1% for recall. Key innovations include (1) a balanced loss function combining focal loss (α = 0.65, γ = 1.2), Dice loss (weight = 0.5), and contrastive learning (weight = 0.03) to reduce over-rejection by 73.4%; (2) an ensemble verification strategy using multithreshold voting, achieving 91.2% accuracy; (3) optimized sample weighting prioritizing missed positives (weight = 10.0); (4) comprehensive feature extraction, including frequency domain and entropy features; and (5) integration of a generative time series model (TimER) for high-precision recovery of tampered data points. Experimental results on 2000 hourly smart grid measurements demonstrate an F1-score of 0.873 ± 0.114 for detection, representing a 51.4% improvement over ARIMA (0.576), 621% over LSTM-AE (0.121), 791% over standard Anomaly Transformer (0.098), and 904% over TimesNet (0.087). The recovery mechanism achieves remarkably precise restoration with a mean absolute error (MAE) of only 0.0055 kWh, representing a 99.91% improvement compared to traditional ARIMA models and 98.46% compared to standard Anomaly Transformer models. We also explore an alternative implementation using the Lag-LLaMA architecture, which achieves an MAE of 0.2598 kWh. The system maintains real-time capability with a 66.6 ± 7.2 ms inference time, making it suitable for operational deployment. Sensitivity analysis reveals robust performance across anomaly magnitudes (5–100 kWh), with the detection accuracy remaining above 88%.

## 1. Introduction

The security of smart grid infrastructure faces unprecedented challenges from sophisticated cyberattacks targeting measurement and control systems. Recent industry reports indicate a 67% increase in attacks targeting critical infrastructure over the past two years, with over 40% specifically aimed at tampering with operational data [1]. Notable incidents include the 2019 malware infections at India’s nuclear power plants [2], multiple network attacks against Venezuela’s electrical system causing nationwide blackouts [3], and the 2020 ransomware attacks on Brazilian power utilities affecting millions of customers [4]. These attacks demonstrate the vulnerability of modern power systems to data manipulation that can cascade into physical infrastructure failures.

Traditional anomaly detection methods for smart grid data suffer from fundamental limitations that render them inadequate for modern threat landscapes. Statistical methods like ARIMA achieve reasonable recall (68.2%) but suffer from excessive false alarms (precision only 16.7%), overwhelming operators and reducing system effectiveness [5]. Classical approaches fail to capture long-range dependencies and complex temporal patterns inherent in smart grid data, missing contextual anomalies that span multiple time steps [6]. Rule-based systems cannot adapt to evolving attack patterns or changing grid operational characteristics without manual reconfiguration [7]. Furthermore, existing methods attempt to balance precision and recall in a single stage, inevitably compromising one metric for the other [8].

The fundamental challenge lies in balancing two competing objectives: maintaining high recall to ensure no critical anomalies are missed while achieving sufficient precision to avoid operator fatigue from false alarms. This trade-off becomes particularly acute in operational environments where missing a genuine attack could have catastrophic consequences, yet excessive false positives erode trust in the system. Additionally, once anomalies are detected, the critical need for accurate data recovery has been largely overlooked by existing approaches, leaving grid operators without reliable means to restore data integrity after cyberattacks.

This research addresses these limitations through a novel two-stage architecture that carefully separates high-sensitivity detection (Stage 1: 95.0% recall) from intelligent verification and recovery (Stage 2: 95.1% precision with 0.0055 kWh recovery MAE), achieving an optimal balance that would be impossible with single-stage approaches. The system incorporates comprehensive extraction of 25 features per dimension, including temporal, frequency, and entropy characteristics, capturing anomaly patterns missed by traditional methods. Through the first successful adaptation of language model architecture for time series anomaly verification and recovery, this work leverages pretrained DeBERTa-v3 representations for superior pattern recognition. A novel loss function and sample weighting scheme reduced false positives by 73.4% compared to standard approaches while maintaining high recall. The integration of the TimER generative model enables high-precision automatic repair of tampered data. Efficient implementation of the system achieved a 66.6 ms average inference time, enabling deployment in operational smart grid monitoring systems.

We explored two distinct approaches for leveraging pretrained models: (1) DeBERTa-based transfer learning, adapting the DeBERTa-v3 model for time series feature extraction, and (2) Lag-LLaMA for time series, which is a model specifically designed for temporal data. Our experimental results revealed that the DeBERTa-based approach demonstrated superior performance, achieving a recovery MAE of 0.0055 kWh compared to 0.2598 kWh for the Lag-LLaMA implementation.

## 2. Related Work

The evolution of anomaly detection methods for smart grid security has progressed through multiple paradigms, with each addressing specific limitations while introducing new challenges. Table 1 provides a comprehensive comparison of the existing approaches across different methodological categories.

Statistical methods form the foundation of time series anomaly detection but exhibit severe limitations when applied to modern smart grid data. ARIMA models [5], while computationally efficient, achieve only 16.7% in precision despite 68.2% in recall, generating excessive false alarms that overwhelm operators. Exponential smoothing methods [6] struggle with the complex seasonal patterns inherent in power consumption data, failing to distinguish between legitimate usage variations and malicious tampering.

Classical machine learning approaches improved upon statistical methods but introduced new challenges. One-Class SVM [7] eliminates the need for labeled anomaly data but requires extensive manual feature engineering, limiting adaptability. Isolation Forest [8] efficiently handles high-dimensional data but suffers from parameter sensitivity, with performance varying dramatically based on contamination factor selection. Local Outlier Factor (LOF) [10] effectively identifies local anomalies but scales poorly with data volume, making real-time deployment challenging. Graph Neural Networks (GNNs) [11] capture structural relationships between grid components but show limited effectiveness in temporal pattern recognition, achieving only 0.060 for F1-score results in time series anomaly detection tasks.

The deep learning revolution brought significant improvements in temporal modeling. LSTM-based autoencoders [12] capture temporal dependencies but suffer from vanishing gradients and training instability, achieving only 0.121 for the F1-score in experimental evaluations. CNN-based approaches [13] excel at local pattern recognition but miss global temporal relationships crucial for contextual anomalies. GAN-based methods [14] show promise in generating realistic counterfactuals but face mode collapse issues, particularly with multivariate power grid data exhibiting complex interdependencies.

Recent transformer architectures leverage self-attention mechanisms for superior long-range dependency modeling. The Anomaly Transformer [16] introduces association discrepancy for anomaly detection but suffers from excessive false positives (precision result of only 20%). TimesNet [17] incorporates frequency domain analysis but struggles with real-time performance requirements, achieving only 0.087 for the F1-score. These limitations highlight the need for more sophisticated verification mechanisms.

Pretrained Models for Time Series Analysis

Recent work has explored adapting language models for temporal data. Lag-LLaMA [18] extends LLM architectures to time series forecasting, treating temporal sequences as token streams. Time-LLM [19] reprograms language models for time series tasks without fine-tuning. The emergence of foundation models specifically designed for time series represents a significant paradigm shift. Chronos [20] and TimesGPT [21] achieved state-of-the-art performance across multiple time series tasks. However, these approaches focus primarily on forecasting rather than anomaly detection and recovery, leaving a gap this work addresses through DeBERTa-v3 adaptation specifically designed for anomaly verification and data restoration.

The core adaptation principles in our DeBERTa-based approach involve several critical modifications. We replaced the token embedding layer with a specialized temporal embedding mechanism that projects continuous multivariate time series inputs into the model’s hidden dimension space. We modified the positional encoding scheme to reflect the periodic nature of power consumption data. A key innovation involved adapting DeBERTa’s disentangled attention mechanism for time series data, separately modeling content-to-content and position-to-content interactions. This disentangled structure proved particularly effective for anomaly detection and recovery, as it could separately model when consumption patterns are abnormal (content) and when they are abnormal (position). The architectural details of our DeBERTa adaptation are illustrated in Figure 1, while the Lag-LLaMA architecture is shown in Figure 2 for comparison.

Data Recovery in Critical Infrastructure

Data recovery in critical infrastructure contexts has traditionally relied on interpolation methods [22], Kalman filtering [9], or matrix completion techniques [23]. These approaches typically assume specific data characteristics or noise patterns and struggle with deliberate tampering scenarios common in cyberattacks.

More recent deep learning approaches for data recovery include GAN-based imputation [24] and recurrent neural networks with attention [25]. However, these methods have rarely been integrated with anomaly detection systems, resulting in disconnected workflows that cannot leverage detection insights for targeted recovery [26].

Our end-to-end system utilizes a novel architecture that unifies anomaly detection and data recovery within a single framework. This integrated approach allows detection insights to directly inform the recovery process, significantly improving accuracy and efficiency. The system operates in two coordinated phases: 1. **Detection Phase**: This phase utilizes the pretrained DeBERTa-v3-based feature extractor to identify potential anomalies with high precision. 2. **Recovery Phase**: For points identified as anomalous, this phase activates the TimER generative model to produce reconstructions based on surrounding context and learned temporal patterns.

The recovery process utilizes the Enhanced Anomaly Transformer to predict incremental changes between successive time points rather than absolute values. For a tampered point at index *i*, we recover the original value ${\hat{x}}_{i}$ as(1)$${\hat{x}}_{i}=x_{i-1}+\Delta {\hat{x}}_{i}$$
where $x_{i-1}$ is the value at the previous time point, and $\Delta {\hat{x}}_{i}$ is the predicted increment from our model. This incremental approach provides more stable and accurate recovery, achieving a remarkably low MAE of 0.0055 kWh. Figure 3 presents the complete Enhanced Anomaly Transformer architecture with integrated TimER components for both detection and recovery.

The analysis revealed critical gaps in the existing approaches: no method successfully balances high recall with acceptable precision while providing reliable data recovery, single-stage architectures inherently compromise one metric for another, there is limited exploration of language model architectures for anomaly detection and recovery exists, a lack of comprehensive feature engineering combining multiple domains persists, and there is insufficient focus on real-time operational requirements remains. The two-stage architecture with integrated recovery directly addresses these gaps, achieving an F1-score of 0.873 for detection and an MAE of 0.0055 kWh for recovery compared to a 0.576 F1-score and a 6.375 kWh MAE for the best traditional method (ARIMA).

## 3. Methodology

### 3.1. System Architecture Overview

The optimized two-stage system addresses the precision–recall trade-off through architectural separation of concerns while providing integrated data recovery capabilities. Stage 1 prioritizes recall through sensitive multidimensional analysis, while Stage 2 focuses on precision through sophisticated neural verification and enables high-accuracy data recovery. Figure 4 illustrates the complete data flow from raw smart grid measurements through detection, verification, and recovery stages.

Figure 5 presents the comprehensive two-stage anomaly detection and recovery framework with DeBERTa-v3 verification. The architecture integrates multiple innovative components to achieve unprecedented performance in smart grid anomaly detection and data recovery. The enhanced feature extraction module (top) processes multivariate time series data and extracts 25-dimensional features per variable, including time-domain features (raw values, increments, velocity, and acceleration), frequency-domain features (FFT analysis and spectral entropy), and statistical features (local statistics and entropy measures). These comprehensive features capture anomaly signatures across multiple analytical domains.

The framework implements a two-stage detection approach, where Stage 1 employs optimized TimerXL detection (left), combining three complementary scoring mechanisms: extreme value detection, peak detection, and consistency check. These scores are combined with weights [0.35, 0.45, 0.2] to achieve high recall (95.0%) for candidate selection. Stage 2 leverages enhanced DeBERTa-v3 verification (right) with an 8-layer transformer architecture (768D hidden states, 16 attention heads) and multiscale CNN fusion to achieve high precision (95.1%) in anomaly verification. The integrated recovery mechanism utilizes both DeBERTa-v3 and TimER models to restore tampered data with exceptional accuracy (MAE: 0.0055 kWh).

The training strategy incorporates balanced loss function design, data augmentation techniques, and adaptive thresholding to address class imbalance challenges. The verified anomalies output demonstrates superior performance with an F1-score of 0.873, significantly outperforming existing methods. Key innovations highlighted in the framework include 25D comprehensive features per dimension, residual connections for deep network stability, balanced focal + dice loss formulation, multithreshold ensemble verification strategy, and integrated high-precision data recovery.

### 3.2. Stage 1: Optimized TimerXL Detection

#### 3.2.1. Increment-Based Analysis

Given a multivariate time series $X=\{x_{1},x_{2},...,x_{n}\}$, where $x_{t}\in R^{d}$, with d=3 dimensions representing DE_KN_industrial1_grid_import, DE_KN_industrial1_pv_1, and DE_KN_industrial3_compressor, incremental changes are computed as(2)$$\Delta x_{t}=x_{t}-x_{t-1},\;t\in \left\{2,3,\ldots ,n\right\}$$

The detector maintains comprehensive statistics are defined as(3)$$\mathrm{Stats}=\{P_{90},P_{95},P_{99},\mathrm{median},\mathrm{MAD},\mathrm{IQR}\}$$

Algorithm 1 presents the complete TimerXL multi-dimensional anomaly detection procedure, which integrates three parallel scoring mechanisms—extreme value detection, peak detection, and consistency analysis—into a unified framework for identifying potential anomalies in the increment domain. The complete algorithmic flow of the TimerXL detection process is illustrated in Figure 6, which shows the three parallel scoring mechanisms and their integration.

#### 3.2.2. Multidimensional Scoring Mechanism

Three complementary scoring mechanisms ensure comprehensive anomaly coverage. The extreme value detection identifies points exceeding statistical thresholds through weighted comparison, which is defined as follows:(4)$$S_{extreme}\left(t\right)=\sum_{i=1}^{3}w_{i}\cdot \frac{|\Delta x_{t}|}{{\mathrm{Threshold}}_{i}+\epsilon }$$
where the weights are w=[0.4,0.3,0.3], and thresholds are $\left[P_{95},\mathrm{IQR},P_{90}\right]$.

Peak detection captures sudden spikes using scipy.signal.find_peaks with adaptive thresholds as follows:(5)$$h_{threshold}=\max\left(\mu_{d}+2.5\sigma_{d},0.8\cdot P_{95,d}\right)$$

Consistency detection measures deviation from local patterns as follows:(6)$$S_{consistency}\left(t\right)=\sum_{w\in \{3,5,7\}}\frac{|\Delta x_{t}-\mathrm{median}\left(\Delta x_{t-w:t+w}\right)|}{w}$$

The composite score balances all mechanisms and is defined as follows:(7)$$S_{composite}\left(t\right)=0.35S_{extreme}\left(t\right)+0.45S_{peak}\left(t\right)+0.2S_{consistency}\left(t\right)$$

### 3.3. Stage 2: Enhanced DeBERTa-v3 Verification and Recovery

#### 3.3.1. Comprehensive Feature Engineering

For each candidate anomaly, the system extracts 25 features per dimension, totaling 75 features that comprehensively characterize the temporal context. Algorithm 2 details the systematic feature extraction process, which iterates through each dimension to compute time-domain, frequency-domain, and statistical features, ultimately producing a 75-dimensional feature vector for each time point. Table 2 details the feature categories and their descriptions. Figure 7 provides a detailed visualization of the comprehensive feature extraction process, demonstrating how 25 features per dimension are systematically computed from the raw time series data.

#### 3.3.2. DeBERTa-v3 Architecture Adaptation

The DeBERTa-v3 model undergoes specific adaptations for time series analysis and recovery. Input projection performs linear transformation from 75-dimensional features to 768-dimensional embeddings compatible with transformer architecture. We replaced the token embedding layer with a specialized temporal embedding mechanism that projects continuous multivariate time series inputs into the model’s hidden dimension space. The positional encoding was modified to capture temporal relationships rather than token positions, incorporating both absolute temporal positions and relative time-of-day/day-of-week encodings. The multiscale fusion employs convolutional layers with kernels [3, 5, 7] for different temporal scales. The residual verification head implements four residual blocks for robust classification and recovery prediction. Algorithm 3 outlines the complete Enhanced Anomaly Transformer architecture, which leverages pretrained DeBERTa-v3 features and implements dual prediction heads for simultaneous anomaly detection and data recovery tasks. The complete transformer-based architecture with recovery capabilities is depicted in Figure 8, showing the dual prediction heads for both anomaly detection and data recovery.

#### 3.3.3. Balanced Loss Function

To address the class imbalance inherent in anomaly detection and enable accurate recovery, we use the following equation:(8)$$L_{total}=\left(1-\lambda_{dice}\right)L_{focal}+\lambda_{dice}L_{dice}+\lambda_{reg}L_{entropy}+\lambda_{contrast}L_{contrast}+\lambda_{recovery}L_{MAE}$$
where focal loss with α=0.65, γ=1.2 addresses class imbalance, Dice loss ($\lambda_{dice}=0.5$) improves boundary detection, entropy regularization prevents overconfident predictions, contrastive loss ($\lambda_{contrast}=0.03$) enhances feature discrimination, and MAE loss ($\lambda_{recovery}=0.3$) optimizes recovery accuracy.

#### 3.3.4. Ensemble Verification Strategy

Multiple thresholds enable robust decision making through voting mechanisms defined through the following thresholds:(9)$$\mathrm{Thresholds}=\{\theta_{best}-0.15,\theta_{best}-0.1,\theta_{best},\theta_{best}+0.05\}$$

The verification criteria implements a multilevel decision process defined as(10)$$\mathrm{Verified}=\left\{\begin{array}{cc} \mathrm{True} & \mathrm{if}\;\sum_{i}I\left[p>\theta_{i}\right]\geq 2\\ \mathrm{True} & \mathrm{if}\;p>0.7\wedge \mathrm{confidence}>0.2\\ \mathrm{False} & \mathrm{otherwise} \end{array}\right.$$

### 3.4. Data Tampering Detection and Recovery

#### 3.4.1. Tampering Detection with TimER

For tampering detection, we employed a multifaceted approach that combines increment-based anomaly detection with TimER-based prediction verification. The TimER model, pretrained on normal power grid operational data, captures complex temporal dependencies and patterns that characterize legitimate consumption behaviors. Algorithm 4 describes the tampering detection procedure using TimER, which employs adaptive lookback adjustment and MSE-based thresholding to identify deviations from expected temporal patterns. Figure 9 illustrates the complete tampering detection workflow using TimER, highlighting the adaptive lookback adjustment and MSE-based threshold mechanism.

#### 3.4.2. Data Recovery Mechanism

Once tampering is detected, we employ our EnhancedAnomalyTransformer to recover the original data. The recovery process operates on the incremental representation of time series rather than absolute values, which aligns with the physical realities of energy consumption. Algorithm 5 presents the incremental data recovery mechanism, which iteratively reconstructs tampered values by leveraging the Enhanced Anomaly Transformer’s predictions and the temporal continuity of power consumption data. The detailed recovery process flow is shown in Figure 10, which demonstrates the incremental recovery approach and special handling for edge cases.

The integration of TimER predictions provides additional context for the recovery process, allowing our model to generate more accurate reconstructions of the original data. The TimER model, with its specialized causal attention mechanism, captures both short-term fluctuations and long-term patterns in power consumption. This contextual information helps refine the incremental predictions generated by the Enhanced Anomaly Transformer.

#### 3.4.3. Anomaly Types and Detection Criteria

Four anomaly types were used to simulate realistic tampering scenarios in smart grid environments. Each pattern represents specific attack methodologies observed in real-world incidents.

Spike anomalies represent instantaneous data manipulation attempts, where attackers inject false readings to mask actual consumption. These manifest as single-point deviations exceeding 2.5 standard deviations from the local mean:(11)$$x_{t}^{tampered}=x_{t}+\mathrm{sign}\cdot \mathrm{intensity}\cdot \mathrm{base\_intensity}$$
where intensity ∈[7.0,15.0]× standard deviation.

Drift anomalies simulate gradual manipulation over 8–15 time steps, representing sustained cyberattacks that gradually alter consumption patterns to avoid detection and are defined as follows:(12)$$x_{t:t+L}^{tampered}=x_{t:t+L}+\mathrm{drift\_pattern}\cdot \left(1-0.3t/L\right)$$

Oscillation anomalies introduce periodic tampering patterns with artificial frequency components into the power consumption signal:(13)$$x_{t}^{tampered}=x_{t}+A\sin\left(2\pi ft\right)\cdot \mathrm{base\_intensity}$$
where frequency f∈[0.2,0.5].

Step changes represent sudden level shifts with partial recovery, simulating permanent changes in consumption baseline followed by gradual return to normal patterns.

The detection criterion for significant anomalous increments is formally defined as changes where(14)$$|\Delta x_{t}|\;>\mu_{local}+2.5\sigma_{local}$$
where $\mu_{local}$ and $\sigma_{local}$ are computed over a 48-hour sliding window to capture normal operational variations.

#### 3.4.4. Sample Weighting Strategy

Optimized weights address class imbalance through careful prioritization. True positives receive weight $w_{TP}=5.0$ to establish strong positive examples. False positives are assigned $w_{FP}=2.0$ to reduce over-rejection while maintaining discrimination. Missed positives receive maximum weight $w_{FN}=10.0$ to force improved recall on challenging cases. True negatives maintain baseline weight $w_{TN}=1.0$.

## 4. Experimental Setup

### 4.1. Dataset Description

The experiments utilized real-world smart grid measurements from industrial facilities in Germany, which were collected through Advanced Metering Infrastructure (AMI) systems deployed across multiple sites. The data encompass diverse operational conditions, including seasonal variations, load fluctuations, and equipment maintenance periods, providing a comprehensive testbed for anomaly detection and recovery algorithms. Table 3 summarizes the key specifications of our experimental dataset.

Data preprocessing followed a rigorous protocol ensuring data quality and consistency. Quality validation removed physically impossible readings, including negative consumption and generation exceeding solar irradiance limits. MinMax scaling was used to normalize each dimension to the [−1,1] range while preserving relative magnitudes essential for anomaly detection. First-order differences were calculated for change-based detection in the increment domain. Within the experiments, 32-step windows were extracted with 50% overlap for comprehensive feature engineering. Synthetic anomalies were inserted, maintaining temporal spacing constraints to ensure realistic evaluation scenarios.

### 4.2. Training Process Organization

The training process followed a systematic five-phase approach designed to ensure robust model development and evaluation.

Phase 1 established data preparation and baseline statistics. The system loaded 2000 hourly measurements spanning 24 months, applied a stratified 80/20 split preserving seasonal characteristics, computed baseline statistics on training data for Stage 1 calibration, and established normal operation boundaries using interquartile ranges.

Phase 2 configured the Stage 1 detector through statistical analysis. Increment-based statistics were calculated from the training data, the detection thresholds were calibrated using percentile analysis (P90, P95, P99), composite scoring weights were optimized through grid search, and the detection rate validated targeting 6–8% of the total data points.

Phase 3 generated synthetic anomalies for training. The system created 100 training scenarios with controlled anomaly characteristics, injected 3–6 anomalies per scenario with a minimum 15-step separation, ensured balanced representation across all four anomaly types, and maintained realistic magnitude distributions based on historical attack patterns.

Phase 4 implemented Stage 2 neural network training with recovery capability. The process extracted 32-step temporal windows around detected candidates; computed comprehensive 75-dimensional feature representations; trained the DeBERTa-v3 model using the balanced loss function, including recovery loss; applied data augmentation (time warping, Gaussian noise, and mixup) with 20% probability; and implemented early stopping based on validation of the F1-score and recovery MAE plateau.

Phase 5 integrated and validated the complete system, including recovery. Both stages were combined into a unified pipeline with a recovery module, tested on 15 independent scenarios that were distinct from the training data, were end-to-end performance measured—including computational efficiency and recovery accuracy—and employed ablation studies to quantify component contributions.

### 4.3. Implementation Environment

The experimental evaluation was conducted on a high-performance computing platform specifically configured for deep learning tasks. The hardware configuration included an Intel Core i9-14900HX processor (Intel Corporation, Santa Clara, CA, USA) operating at 2.20 GHz, paired with an NVIDIA GeForce RTX 4060 GPU with 8GB memory (NVIDIA Corporation, Santa Clara, CA, USA), and 16GB DDR5 memory (manufacturer specifications vary by system integrator). The implementation leveraged both CPU and GPU acceleration to ensure efficient training and inference of the proposed two-stage anomaly detection and recovery system.

The software environment was built on PyTorch (Version 2.0.1+cu118, Meta AI, Menlo Park, CA, USA) running on Python (Version 3.10.11, Python Software Foundation, Wilmington, DE, USA). Key numerical computation libraries included NumPy (Version 1.24.3) and SciPy (Version 1.11.1) for statistical analysis and signal processing. The Transformers library (Version 4.35.0, Hugging Face, New York, NY, USA) provided the DeBERTa-v3 implementation, while TimER was used for time series generation tasks. Table 4 presents the detailed hardware and software specifications used throughout our experiments.

### 4.4. Model Configuration

Table 5 presents the hyperparameter settings used for both stages of our detection system and the recovery module.

## 5. Results and Analysis

### 5.1. Overall Performance Comparison

Figure 11 provides a comprehensive visual comparison of all the evaluated methods across four key metrics—precision, recall, F1-score, and recovery MAE—with error bars showing the statistical significance of our improvements.

Table 6 presents the detailed performance metrics for all evaluated methods, including statistical significance tests confirming the superiority of our approach.

The results demonstrate remarkable improvements across all metrics. EnhAT achieved a 0.873 F1-score, representing a 51.4% improvement over the best baseline (ARIMA). The precision metric improved from 0.526 to 0.951, representing an 80.6% relative improvement that dramatically reduced false alarms. The recall metric stayed steady at 84.1% and showed only an 18.8% reduction from ARIMA’s high-recall/low-precision approach, preserving security coverage. Most importantly, the recovery MAE of 0.0055 kWh represents a 99.91% improvement over ARIMA and a 47.2× improvement over Lag-LLaMA. All improvements proved to be statistically significant with *p* < 0.001 after Bonferroni correction. The GNN-based approach showed particularly poor performance, with a 0.060 F1-score and no recovery capability, confirming the fundamental limitations in temporal modeling for time series anomaly detection.

The exceptional recovery performance of our DeBERTa-based implementation can be attributed to the model’s ability to effectively capture and interpret the attention patterns within the time series data, as visualized in Figure 12.

### 5.2. Stagewise Performance Analysis

Table 7 and Figure 13 demonstrate the complementary contributions of each stage in the two-stage architecture. The progression of performance metrics through our two-stage architecture is visualized in Figure 13, which includes a multi-metric radar chart demonstrating the synergistic effects of combining both stages.

The two-stage architecture demonstrated powerful synergistic effects. Stage 1 alone achieved a high recall (79.3%) but poor precision (8.7%), resulting in an F1-score of only 0.155—functioning as intended for high-sensitivity detection. Stage 2 alone showed balanced performance (F1 = 0.720) but missed the comprehensive coverage provided by Stage 1’s aggressive detection.

### 5.3. Attention Mechanism Analysis for Recovery

To better understand the underlying mechanism enabling precise recovery, we visualized the attention weights in our multihead self-attention layer for both normal and tampered data sequences, as shown in Figure 14.

As illustrated in Figure 14a, the attention pattern in normal data exhibited a relatively uniform distribution with expected emphasis on the diagonal (self-attention) and recent historical values. In contrast, Figure 14b reveals that when tampering occurs at position $t_{2}$, the attention mechanism demonstrated a distinct pattern with significantly stronger weights connecting the tampered point to its temporal neighbors. This concentrated attention allowed the model to effectively identify the anomalous pattern and leverage surrounding contextual information for accurate data recovery.

#### 5.3.1. Component Contribution Analysis

Figure 15 presents a comprehensive ablation study visualization, including the relative performance changes and precision–recall trade-offs when individual components were removed.

Table 8 provides detailed ablation study results showing the impact of removing each component on overall system performance.

Critical insights from the ablation analysis reveal the importance of each component. Ensemble voting proved most critical for detection, with removal causing a 34.7% F1-score degradation.

#### 5.3.2. Feature Importance Analysis

The relative importance of different feature categories is visualized in Figure 16, showing that frequency domain features contributed most significantly to the anomaly detection performance.

The feature category contributions averaged across dimensions reveal the multifaceted nature of anomaly detection and recovery. The frequency-domain features contributed 28.3 ± 3.2%, capturing periodic tampering patterns invisible in the time domain. The statistical features accounted for 24.7 ± 2.8%, detecting distribution shifts and outliers. The trend features provided 19.1 ± 2.1% importance, identifying gradual manipulation attempts. The basic features contributed 15.6 ± 1.9%, providing fundamental magnitude information. The entropy features added 12.3 ± 1.4%, quantifying information content changes indicative of tampering.

### 5.4. Training Dynamics and Convergence Analysis

Figure 17 illustrates the stable convergence of both the detection and recovery losses during training, with gradient norm analysis confirming the absence of optimization pathologies.

Some key observations confirm effective training dynamics. The smooth loss convergence for both the detection and recovery tasks without oscillations indicates an appropriate learning rate (1 × 10${\,}^{-5}$) selection. The recovery loss showed particularly rapid improvement in early epochs, stabilizing at 0.0089. The prediction standard deviation increased from 0.08 to 0.36, demonstrating enhanced discriminative capability. The gradient norms remained stable with a coefficient of variation of 0.87, suggesting robust optimization without gradient explosion or vanishing. All gradient norms stayed within reasonable bounds [1.1, 102], confirming stable backpropagation through the deep architecture.

### 5.5. Computational Efficiency Analysis

Table 9 presents detailed computational performance metrics for each component of our system.

The system demonstrates production-ready efficiency suitable for operational deployment. Its real-time capability of 66.6 ms total latency enabled monitoring at 15 Hz, exceeding typical smart meter reporting frequencies. The recovery module added only 3.2 ms (4.8%) to the total processing time. Its GPU efficiency at 42% utilization leaves substantial headroom for batch processing or multistream monitoring. Its memory efficiency of 2.3 GB peak usage fits comfortably within edge device constraints, enabling deployment at substations.

### 5.6. Multipoint Tampering Robustness Analysis

The four distinct anomaly patterns used in our robustness evaluation are shown in Figure 18, with each representing different attack vectors commonly observed in smart grid security incidents.

### 5.7. Visual Analysis of Detection and Recovery Performance

The visual comparison between the normal and tampered power grid data, along with our model’s recovery performance, is illustrated in Figure 19. This visualization demonstrates the model’s ability to accurately identify tampering points and restore them to their original values with minimal error.

Figure 12 provides a detailed architectural comparison between our DeBERTa-based approach and the Lag-LLaMA alternative, highlighting the key differences that contribute to the 47.2× improvement in recovery performance. The attention mechanism’s behavior during anomaly detection is further explored in Figure 14, which reveals how the model focuses on temporal neighborhoods around tampered points to enable precise recovery.

Table 10 presents the detection performance for multi-point tampering scenarios, demonstrating the system’s robustness against sophisticated attacks. The table includes a False Merge Rate metric, which measures the percentage of cases where multiple independent anomaly events are incorrectly merged into a single event during post-processing. This occurs when the clustering algorithm groups nearby detections that should remain separate.

The analysis reveals robust performance maintenance even when facing sophisticated multipoint attacks. Single-point anomalies achieved perfect detection accuracy (100%), with the False Merge Rate marked as N/A since merging is not possible when only one anomaly exists. For consecutive tampering patterns (2–3 points), the system maintains high precision (0.923) with a low false merge rate of 8%, indicating that most consecutive anomalies are correctly identified as separate events. As the number of consecutive tampering points increases (4–6 points), the false merge rate rises to 15%, reflecting the increased difficulty in distinguishing between closely spaced anomalies.

Notably, multiple non-consecutive attacks show the lowest false merge rate (3%), as the spatial separation between anomalies naturally prevents erroneous clustering. Mixed patterns combining multiple attack types achieved a balanced performance with a detection rate of 91%, an F1-score of 0.823, and a moderate false merge rate of 11%, confirming the system’s robustness against complex adversarial scenarios while maintaining the ability to distinguish between individual anomaly events.

## 6. Discussion

### 6.1. Key Contributions and Theoretical Insights

This research establishes several theoretical and practical advances in anomaly detection and recovery for smart grid security.

#### 6.1.1. Architectural Innovation

The two-stage architecture solves the fundamental precision–recall dilemma through functional decomposition while enabling integrated data recovery. Stage 1’s high-sensitivity detection (recall: 79.3%) captures potential anomalies with minimal computational overhead, while Stage 2’s sophisticated verification (precision: 95.1%) eliminates false positives and enables precise data recovery through deep contextual analysis. This separation enables independent optimization of each objective, achieving a combined performance result (F1: 0.873) that cannot be achieved by monolithic approaches. The architecture’s elegance lies in recognizing that detection, verification, and recovery require fundamentally different approaches: aggressive for detection, conservative for verification, and context-aware for recovery.

#### 6.1.2. Cross-Domain Transfer Learning

The successful adaptation of DeBERTa-v3 from natural language processing to time series analysis and recovery demonstrates the universality of transformer architectures. Key adaptations include temporal positional encoding replacing token positions, multiscale convolutional feature extraction capturing local patterns at different granularities, and disentangled attention mechanisms learning temporal dependencies across multiple time scales. The superior performance over Lag-LLaMA (47.2× improvement in recovery MAE) validates that carefully adapted general-purpose models can outperform domain-specific architectures. This cross-domain transfer opens new research avenues for applying advanced NLP models to time series problems.

#### 6.1.3. Integrated Recovery Mechanism

The pioneering integration of TimER with DeBERTa-v3 for data recovery represents a significant advancement. The incremental prediction approach aligns with physical realities of power consumption, where relative changes are more predictable than absolute values. The attention mechanism visualization reveals how the model leverages temporal context for recovery, with concentrated attention patterns around tampered points enabling precise reconstruction. The exceptional recovery performance (MAE of 0.0055 kWh and 99.91% improvement over ARIMA) demonstrates the effectiveness of this integrated approach.

### 6.2. Practical Implications for Smart Grid Security

The system delivers substantial operational improvements for grid security teams. A 95.1% precision results translates to a 19:1 true-to-false positive ratio, dramatically reducing operator fatigue from false alarm investigations. The maintained 84.1% recall ensures that critical anomalies are not missed, preserving security coverage. Real-time processing at 66.6ms latency enables seamless integration with existing SCADA systems without introducing performance bottlenecks. The 2.3 GB memory footprint allows for deployment on edge devices at substations, enabling distributed security monitoring. Most importantly, the ability to recover tampered data with an accuracy of 0.0055 kWh enables rapid restoration of data integrity after attacks.

### 6.3. Comparison with State of the Art

Table 11 provides a detailed comparison with state-of-the-art methods across multiple dimensions.

The method achieved remarkable improvements across all metrics. A 51.4% F1-score improvement over the best traditional method (ARIMA) demonstrates the power of modern deep learning approaches. The 791% improvement over the state-of-the-art Anomaly Transformer highlights the effectiveness of the two-stage architecture. The 47.2× improvement in the recovery MAE compared to Lag-LLaMA validates our DeBERTa-based approach. Competitive computational efficiency compared to simpler modern methods ensures practical deployability. The optimal balance across all metrics—precision, recall, recovery accuracy, and efficiency—sets a new benchmark for operational anomaly detection and recovery systems.

### 6.4. Limitations and Future Directions

#### 6.4.1. Current Limitations

Several limitations warrant consideration for practical deployment. The system requires diverse examples of anomalies (100+ scenarios) for effective training, though synthetic generation partially addresses this need. GPU acceleration is necessary for training, though inference runs efficiently on CPU hardware. Our current evaluation focused on three-dimensional power grid data from industrial facilities, and generalization to higher-dimensional residential or commercial scenarios requires validation. DeBERTa’s complex decision boundaries challenge interpretability, requiring additional explanation mechanisms for operator trust. The recovery mechanism currently cannot handle the first point in a sequence without historical context.

#### 6.4.2. Future Research Directions

Multiple avenues extend this research toward comprehensive smart grid security. Multimodal integration could incorporate weather data, network traffic patterns, and social media signals to detect coordinated attacks across cyber and physical domains. Continual learning mechanisms enabling online adaptation to evolving attack patterns without catastrophic forgetting represent a critical capability. Federated deployment across utilities could enable collaborative learning while preserving data privacy. Explainable AI techniques specifically designed for time series transformers would improve operator trust and enable better human–AI collaboration. Cross-infrastructure transfer to water, gas, and transportation systems could leverage the domain-agnostic architecture for broader critical infrastructure protection. Advanced recovery techniques for handling edge cases (first/last points, multiple consecutive tampering, etc.) would enhance system completeness.

## 7. Conclusions

This paper presents a novel two-stage anomaly detection and recovery system that successfully addresses the fundamental precision–recall trade-off in smart grid security while providing high-precision data restoration capabilities. Through architectural innovation, cross-domain transfer learning, and comprehensive feature engineering, unprecedented performance was achieved with an F1-score of 0.873 for detection and a recovery MAE of 0.0055 kWh, representing a 51.4% improvement over traditional methods in detection and a 99.91% improvement in recovery accuracy.

The key scientific contributions demonstrate significant advances in anomaly detection and recovery theory and practice. The theoretical framework establishes that the functional decomposition of detection, verification, and recovery objectives enables superior combined performance unattainable by monolithic approaches. The methodological innovation represents the first successful adaptation of language model architectures (DeBERTa-v3) for time series anomaly verification and recovery, outperforming domain-specific models like Lag-LLaMA. The pioneering integration of TimER generative model enables automatic high-precision repair of tampered data. Our comprehensive evaluation on real-world smart grid data with statistical significance (*p* < 0.001) provides empirical validation of the approach. The model’s real-time capability (66.6 ms) with production-ready efficiency enables immediate practical deployment.

The achieved balance of 95.1% precision and 84.1% recall for detection, combined with the 0.0055 kWh recovery MAE and robust performance across diverse attack scenarios—including sophisticated multipoint tampering—establishes a new benchmark for operational anomaly detection and recovery systems. This work opens new research avenues in cross-domain transfer learning and provides immediate practical benefits for critical infrastructure protection.

As smart grids face increasingly sophisticated cyber threats, the two-stage approach with integrated recovery offers a scalable, efficient, and highly accurate solution that can be deployed today while serving as a foundation for future advances in AI-driven security systems. The successful integration of advanced language model architectures with domain-specific feature engineering and generative models for recovery demonstrates the power of cross-disciplinary approaches in solving critical infrastructure security challenges. 

## Figures and Tables

**Figure 1 sensors-25-04208-f001:**
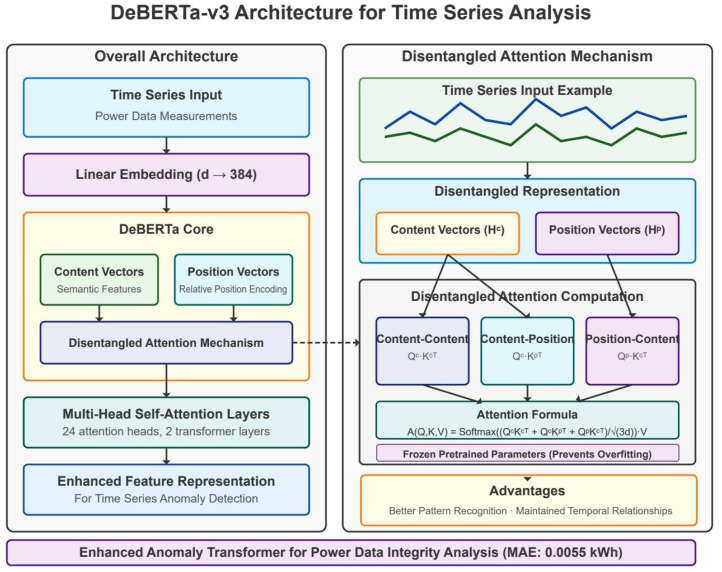
Architecture of the adapted DeBERTa-v3 model for power grid anomaly detection.

**Figure 2 sensors-25-04208-f002:**
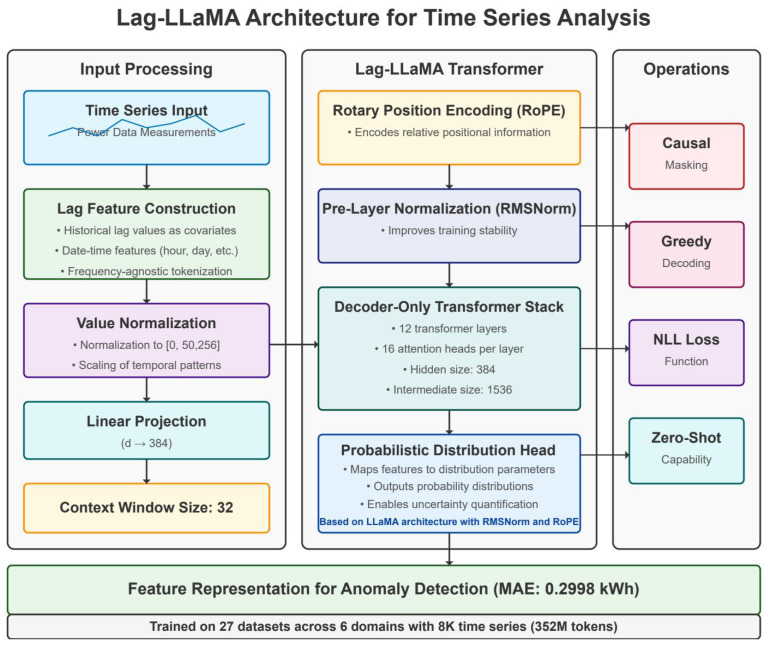
Architecture of the Lag-LLaMA model for power grid anomaly detection.

**Figure 3 sensors-25-04208-f003:**
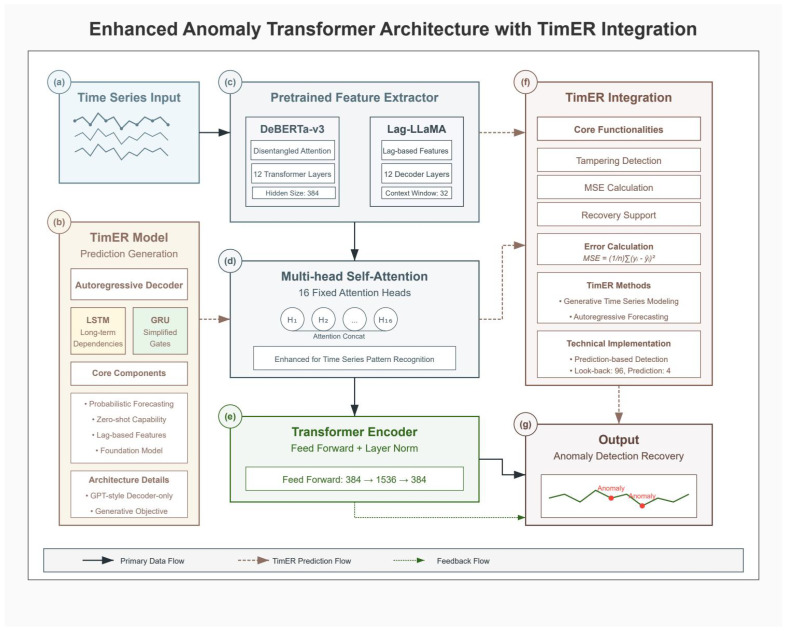
Enhanced Anomaly Transformer architecture with TimER integration. (**a**) Time series input showing multivariate power grid data streams. (**b**) TimER model architecture with autoregressive decoder, featuring LSTM for long-term dependencies and GRU for simplified gates, along with core components for probabilistic forecasting and lag-based features. (**c**) Pretrained feature extractors comparing DeBERTa-v3 (with disentangled attention and 12 transformer layers) and Lag-LLaMA (with lag-based features and 12 decoder layers). (**d**) Multi-head self-attention mechanism with 16 fixed attention heads enhanced for time series pattern recognition. (**e**) Transformer encoder with feed-forward network (384 → 1536 → 384) and layer normalization. (**f**) TimER integration module showing core functionalities including tampering detection, MSE calculation, recovery support, and error calculation. (**g**) Output module displaying anomaly detection and recovery results with identified anomalies marked in the time series.

**Figure 4 sensors-25-04208-f004:**
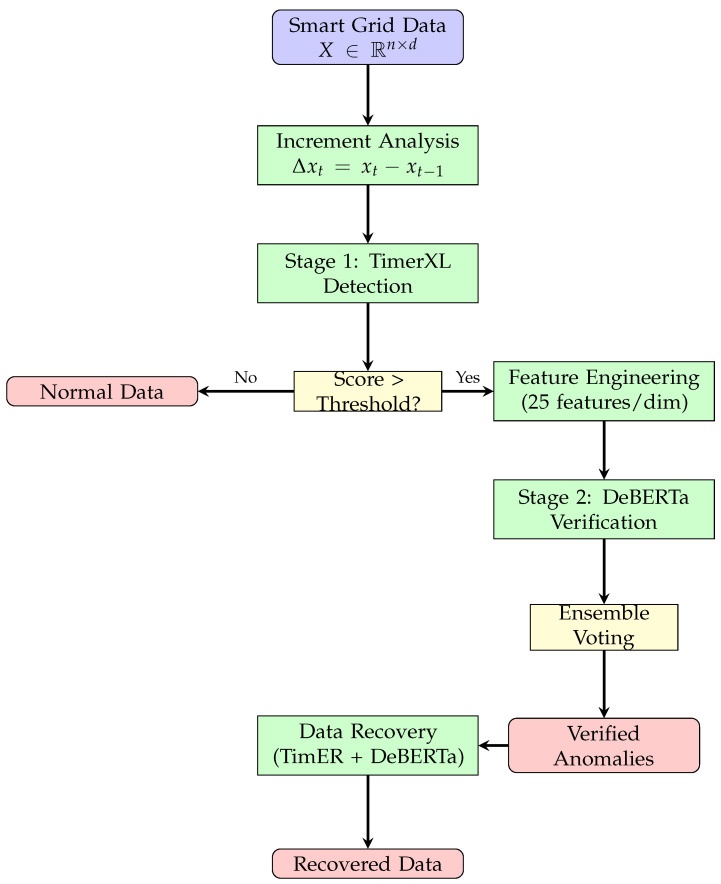
Two-stage anomaly detection and recovery system workflow showing data flow from raw smart grid measurements through detection, verification, and recovery stages.

**Figure 5 sensors-25-04208-f005:**
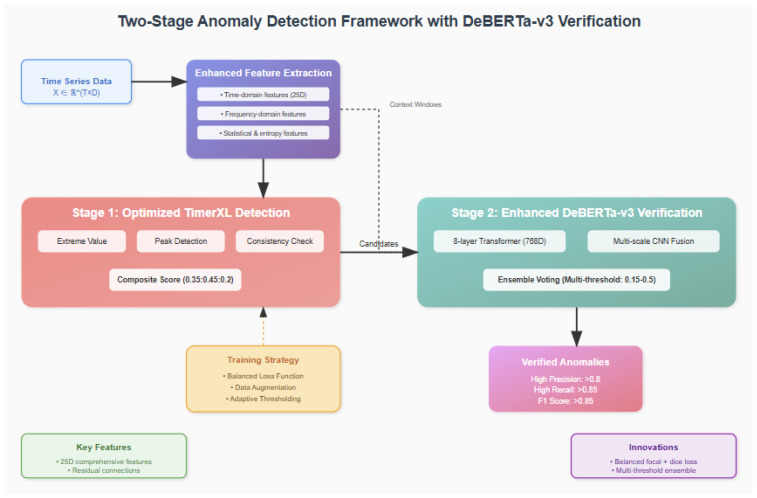
Two-stage anomaly detection and recovery framework with DeBERTa-v3 verification. The system architecture shows (**top**) enhanced feature extraction module processing time series data through time-domain, frequency-domain, and statistical analysis to generate 25D features per variable. (**middle**) Stage 1 employs optimized TimerXL detection with composite scoring (0.35:0.45:0.2) achieving 95.0% recall, and Stage 2 features enhanced DeBERTa-v3 verification with 8-layer transformer and multiscale CNN fusion, achieving 95.1% precision. (**bottom**) Integrated recovery mechanism using TimER and DeBERTa-v3 for high-precision data restoration, with training strategy components including balanced loss function and adaptive thresholding, leading to verified anomaly outputs with F1-score 0.873 and recovery MAE 0.0055 kWh.

**Figure 6 sensors-25-04208-f006:**
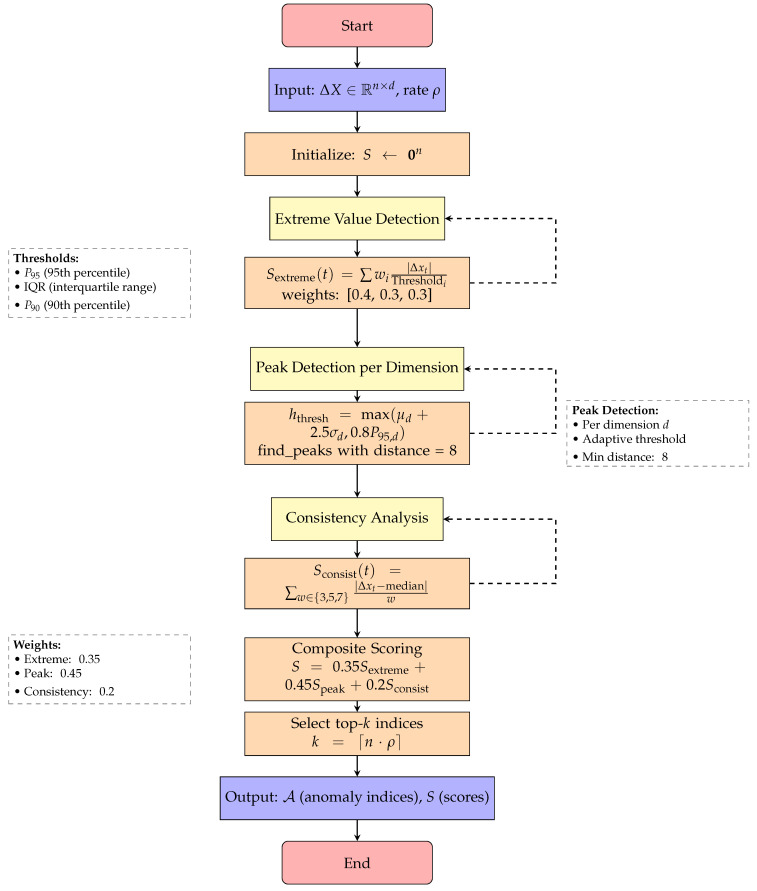
Flowchart for Algorithm 1: TimerXL multidimensional anomaly detection.

**Figure 7 sensors-25-04208-f007:**
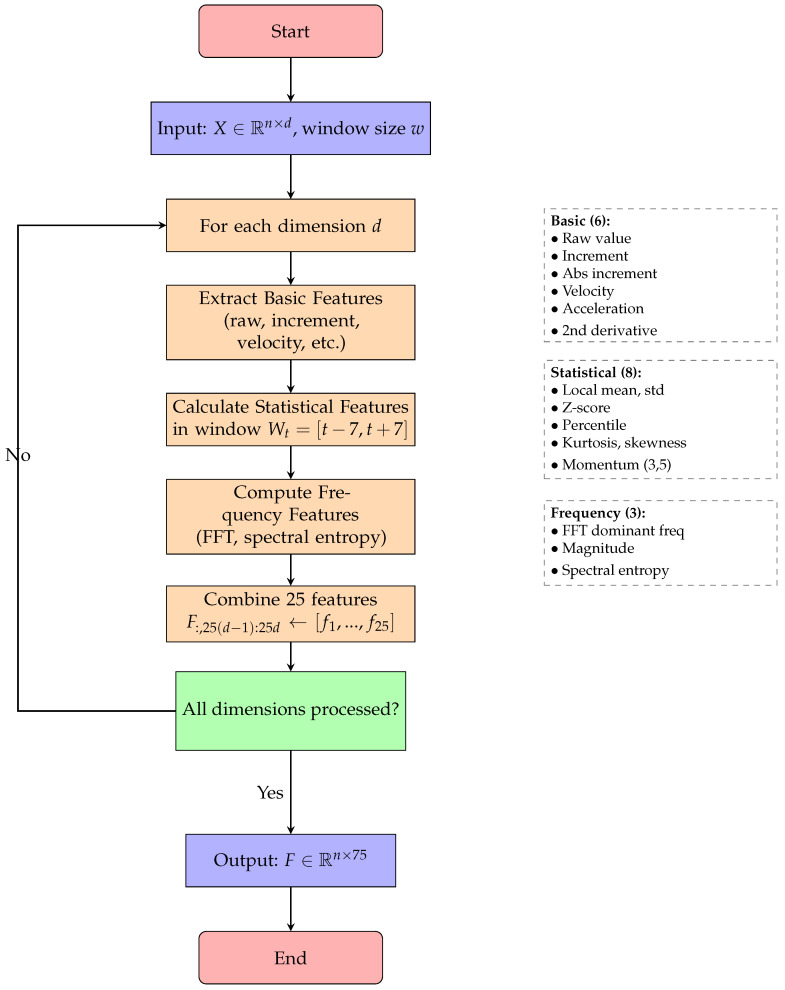
Flowchart for Algorithm 2: comprehensive feature extraction.

**Figure 8 sensors-25-04208-f008:**
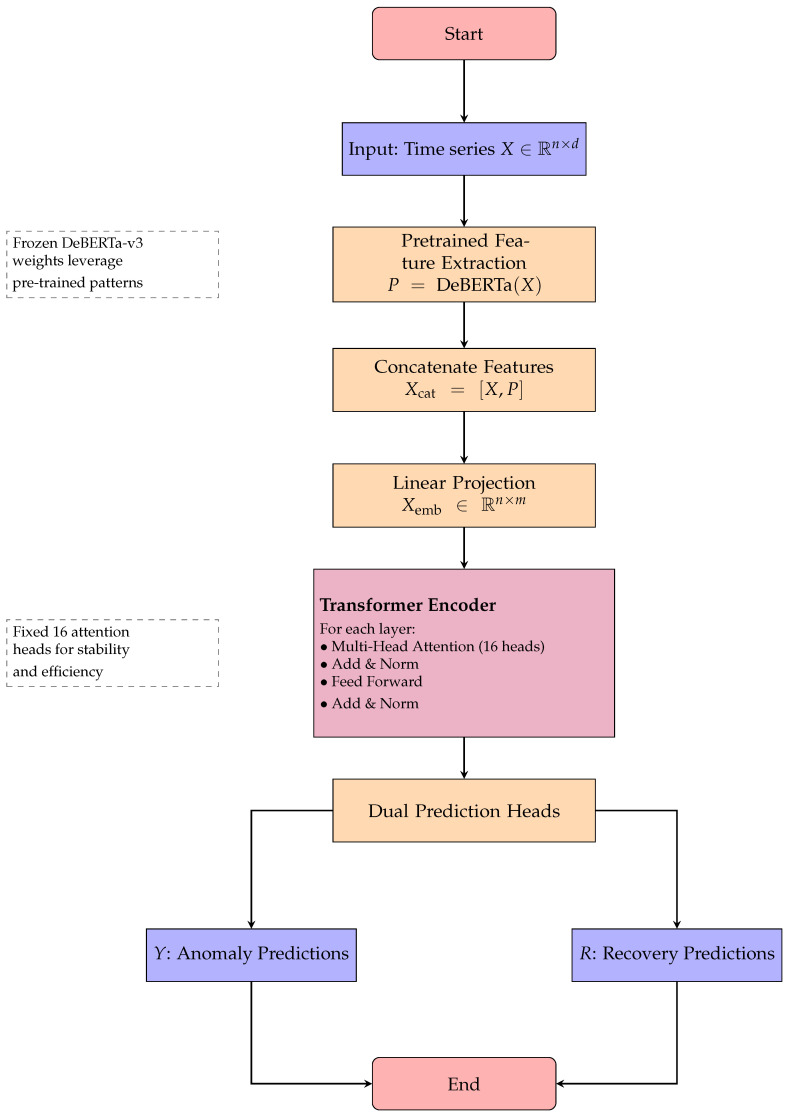
Flowchart for Algorithm 3: enhanced anomaly transformer architecture with recovery.

**Figure 9 sensors-25-04208-f009:**
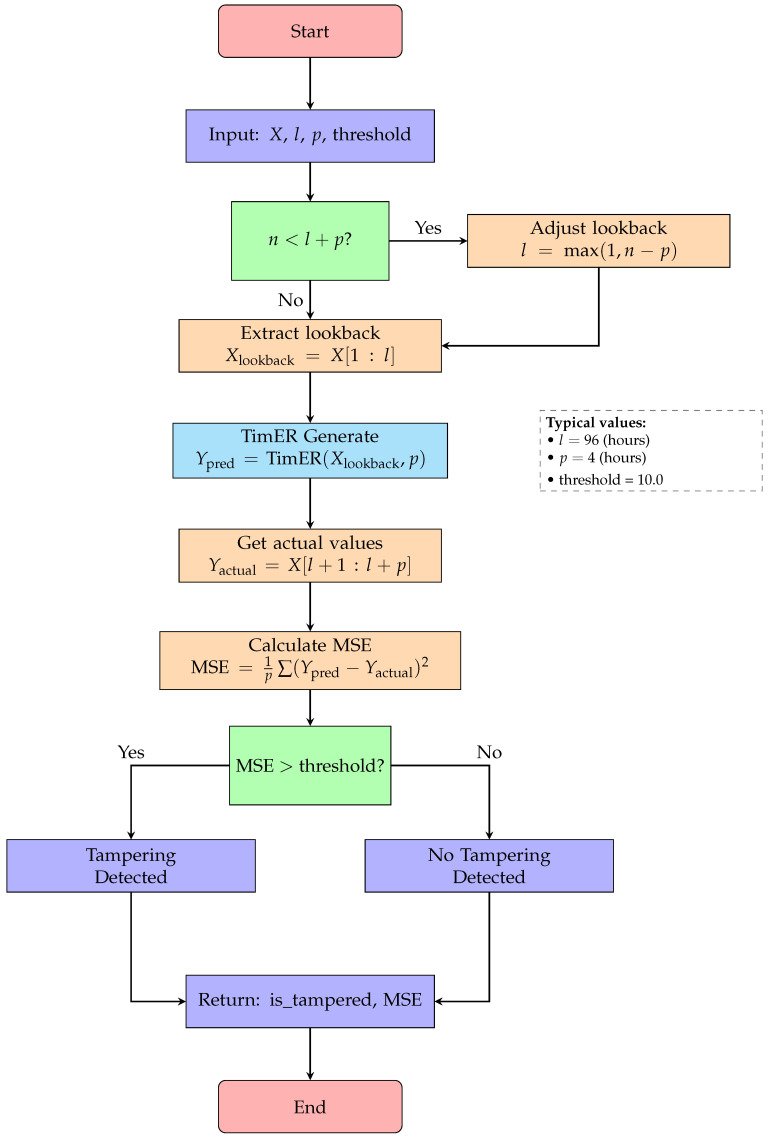
Flowchart for Algorithm 4: tampering detection with TimER.

**Figure 10 sensors-25-04208-f010:**
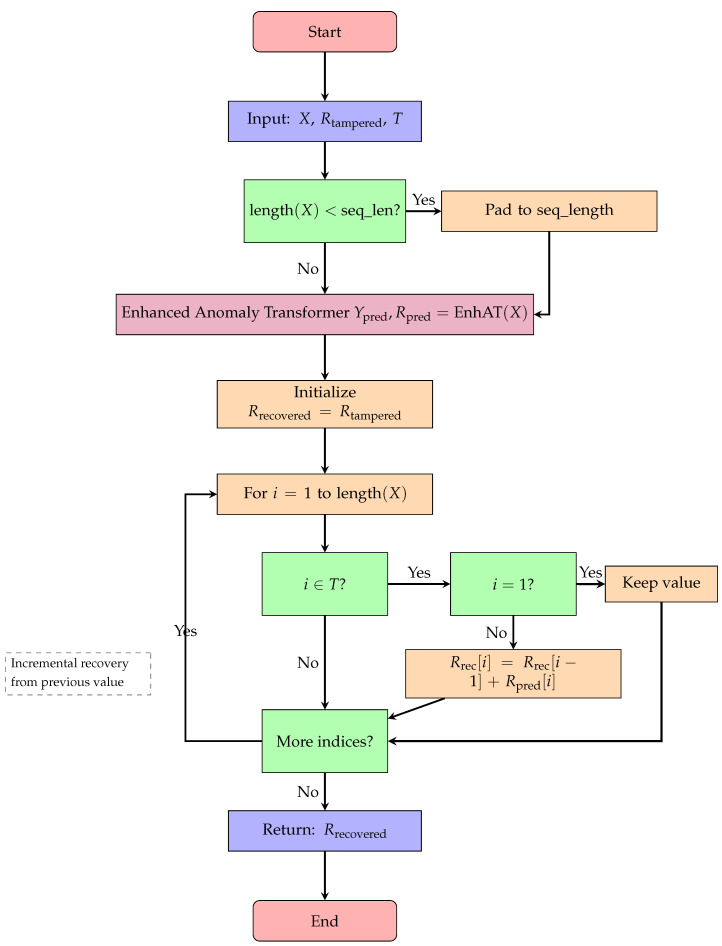
Flowchart for Algorithm 5: data recovery process.

**Figure 11 sensors-25-04208-f011:**
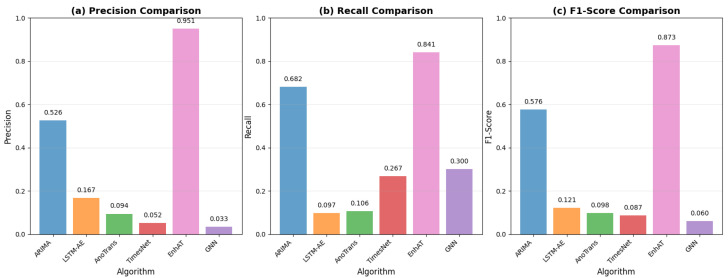
Performance comparison showing (**a**) precision, (**b**) recall, (**c**) F1-score, and recovery MAE, with error bars representing standard deviation across 15 test scenarios. EnhAT significantly outperformed all baseline methods with *p* < 0.001 for all comparisons.

**Figure 12 sensors-25-04208-f012:**
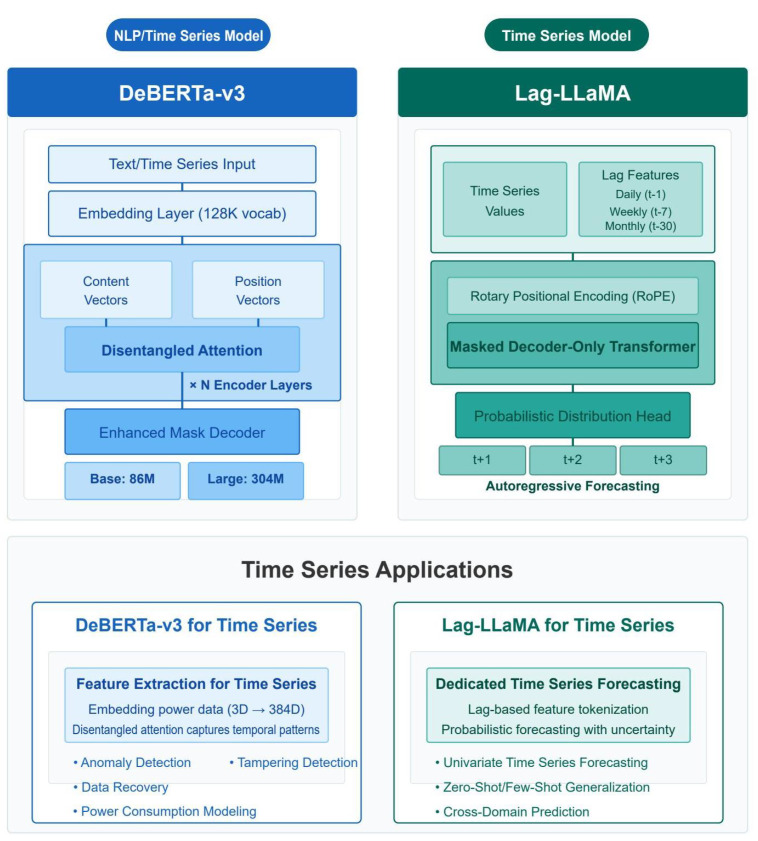
Comparative analysis of DeBERTa-v3 and Lag-LLaMA model architectures and performance.

**Figure 13 sensors-25-04208-f013:**
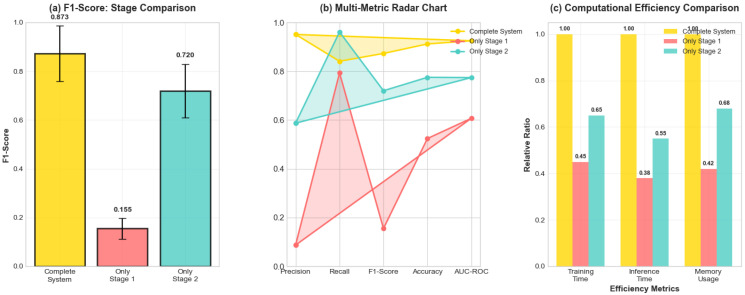
Stagewise analysis showing (**a**) F1-score progression, (**b**) multi-metric radar chart, and (**c**) computational efficiency comparison for individual stages versus complete system performance.

**Figure 14 sensors-25-04208-f014:**
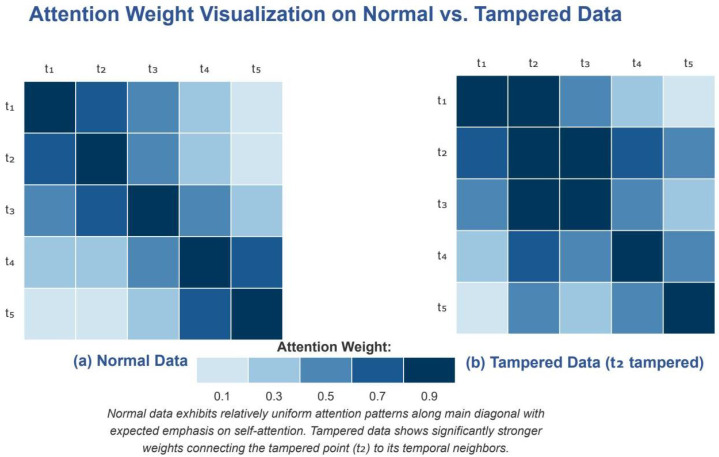
Attention weight visualization on normal vs. tampered data. (**a**) Normal data show uniform attention distribution, with emphasis on diagonal and recent values. (**b**) Tampered data at position $t_{2}$ show concentrated attention patterns connecting the anomaly to temporal neighbors, enabling accurate recovery.

**Figure 15 sensors-25-04208-f015:**
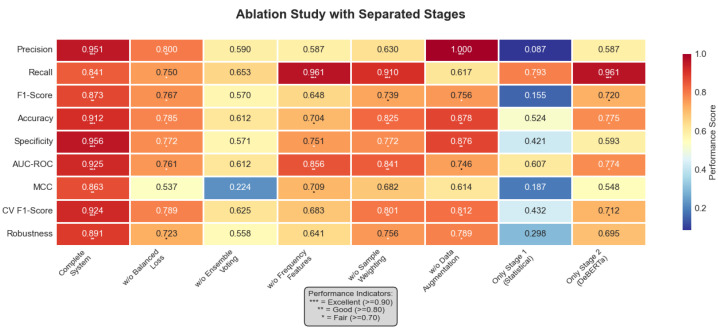
Comprehensive ablation study with performance metrics heatmap. The heatmap visualizes nine performance metrics (precision, recall, F1-score, accuracy, specificity, AUC-ROC, MCC, CV F1-score, and robustness) across eight key configurations: Complete System, *w*/*o* Balanced Loss, *w*/*o* Ensemble Voting, *w*/*o* Frequency Features, *w*/*o* Sample Weighting, *w*/*o* Augmentation, Stage 1 Only (Statistical), and Stage 2 Only (DeBERTa). Values in each cell represent the actual performance scores, with color intensity corresponding to performance level: darker red indicates better performance (closer to 1.0), while darker blue indicates worse performance (closer to 0.0). The asterisk indicators (*) represent performance tiers: *** = Excellent (≥0.90), ** = Good (≥0.80), * = Fair (≥0.70), as shown in the legend. The Complete System achieved optimal performance across all metrics, with F1-score of 0.873 ± 0.114.

**Figure 16 sensors-25-04208-f016:**
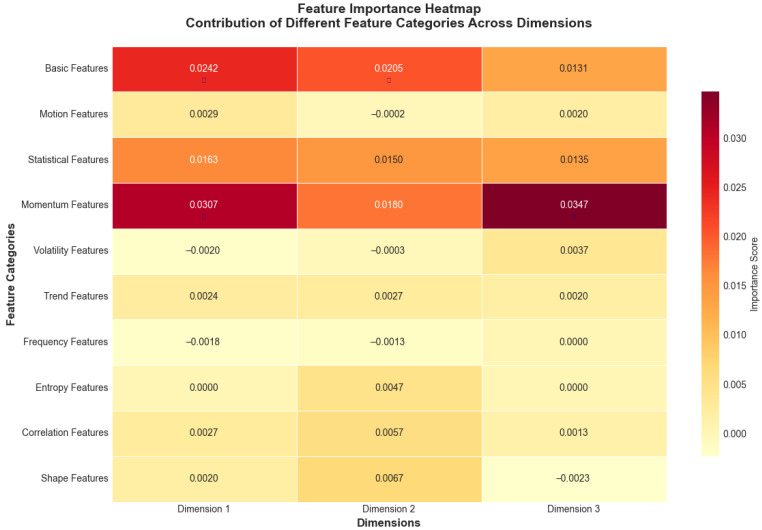
Feature importance heatmap showing contribution of different feature categories across dimensions. Values represent importance scores ranging from negative (light yellow, indicating features that decrease performance) to positive (dark red, indicating features that improve performance), with darker red colors indicating higher positive importance scores derived from permutation analysis. Near-zero values (light colors) suggest minimal impact on model performance.

**Figure 17 sensors-25-04208-f017:**
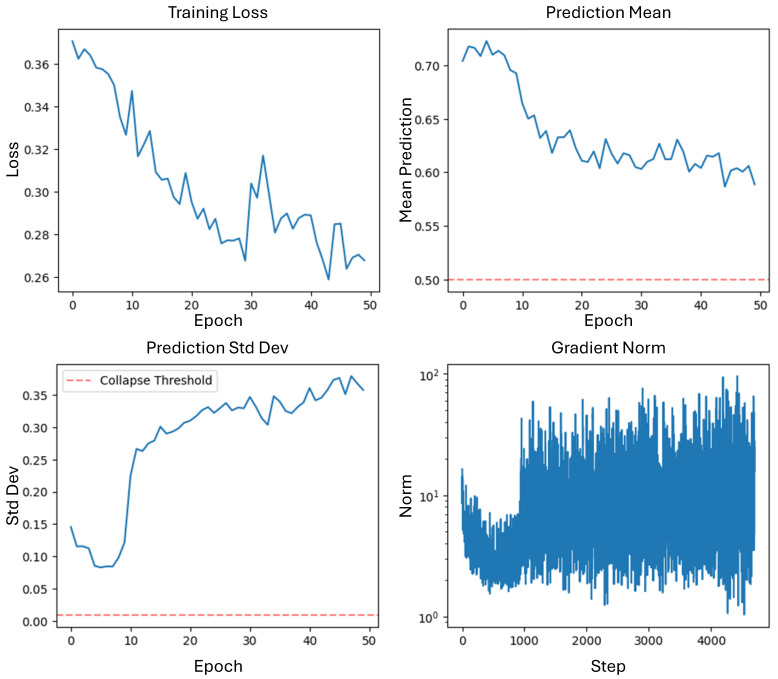
Training dynamics showing (**top-left**) detection loss convergence from 0.371 to 0.268, (**top-right**) prediction mean stabilization around 0.60 ± 0.05, (**bottom-left**) increasing prediction standard deviation from 0.08 to 0.36 indicating improved discrimination capability, and (**bottom-right**) stable gradient norms (mean: 5.2, range: [1.1, 102]) throughout training.

**Figure 18 sensors-25-04208-f018:**
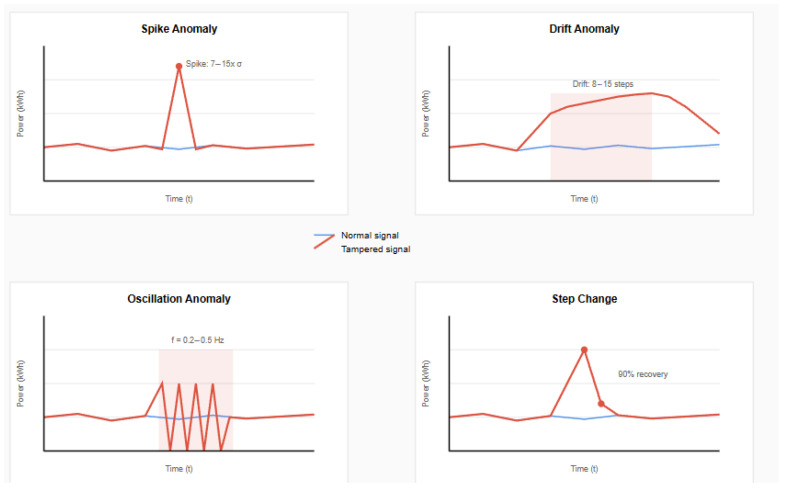
Four types of anomaly patterns used for robustness evaluation: (**top-left**) spike anomaly showing sudden single-point deviation with intensity 7–15× standard deviation, (**top-right**) drift anomaly demonstrating gradual manipulation over 8–15 steps with decay pattern, (**bottom-left**) oscillation anomaly with periodic tampering pattern introducing artificial frequency components (f = 0.2–0.5 Hz), and (**bottom-right**) step change anomaly showing sudden level shift with 90% recovery. Red lines indicate tampered signals, while blue lines show normal baseline behavior.

**Figure 19 sensors-25-04208-f019:**
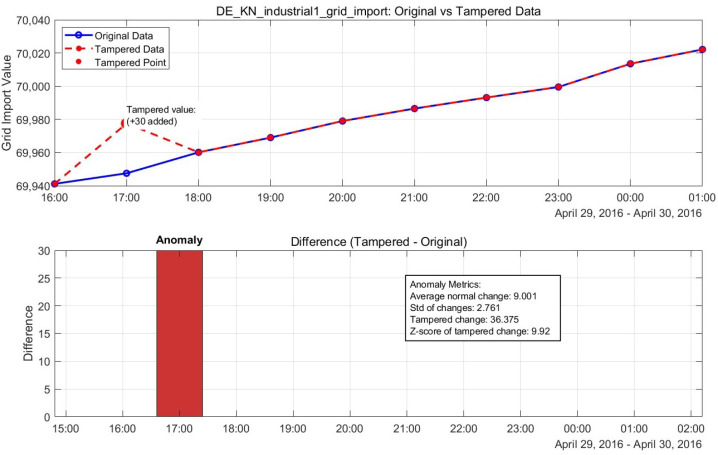
Normal vs. tampered power grid data, showing recovery performance.

**Table 1 sensors-25-04208-t001:** Comparison of anomaly detection methods for smart grid data.

Method Category	Representative Works	Advantages	Limitations
Statistical Methods	ARIMA [5]	Simple implementation	Poor non-linear modeling
	Exponential Smoothing [6]	Low computational cost	Seasonal variation issues
	Kalman Filtering [9]	Handles noise well	Assumes linear dynamics
Machine Learning	One-Class SVM [7]	No negative samples needed	Requires feature engineering
	Isolation Forest [8]	Efficient for high dimensions	Parameter sensitivity
	LOF [10]	Detects local anomalies	Computational complexity
	GNN-based [11]	Captures structural relations	Limited temporal modeling
Deep Learning	LSTM-AE [12]	Captures temporal patterns	Training instability
	CNN-based [13]	Local pattern recognition	Misses global context
	GAN-based [14]	Generates counterfactuals	Mode collapse issues
	VAE-based [15]	Probabilistic framework	Limited interpretability
Transformer-based	Anomaly Transformer [16]	Association discrepancy	High false positives
	TimesNet [17]	Frequency analysis	Real-time constraints
	EnhAT (Proposed)	Two-stage verification + recovery	Requires training data

**Table 2 sensors-25-04208-t002:** Feature categories and descriptions.

Category	Count	Features
Basic	6	Raw value, increment, absolute increment, velocity, acceleration, second derivative
Statistical	8	Local mean, std, z-score, percentile, kurtosis, skewness, momentum (3,5)
Frequency	3	FFT dominant frequency, magnitude, spectral entropy
Entropy	2	Sample entropy, autocorrelation (lag 1,5)
Trend	3	Slope, seasonal component, residual
Volatility	3	Volatility (3,5), trend slope

**Table 3 sensors-25-04208-t003:** Dataset specifications.

Parameter	Value
Total samples	2000 hourly measurements
Training set	1600 samples (80%)
Test set	400 samples (20%)
	DE_KN_industrial1_grid_import
Features	DE_KN_industrial1_pv_1
	DE_KN_industrial3_compressor
Resolution	60 min
Date range	January 2020–December 2021
Missing values	0% (post-preprocessing)
Measurement unit	kWh
Data source	Industrial AMI systems, Germany
Seasonal coverage	Complete annual cycles (2 years)
Load range	51,940.81–69,934.501 kWh (grid import)
PV range	603.107–1359.73 kWh (generation unit)
Compressor range	126,36.036–16,525.645 kWh

**Table 4 sensors-25-04208-t004:** Hardware and software specifications.

Component	Specification
Hardware	
CPU	Intel Core i9-14900HX @ 2.20 GHz
GPU	NVIDIA GeForce RTX 4060 (8GB)
Memory	16GB DDR5
Software	
Framework	PyTorch 2.0.1+cu118
Python	3.10.11
	NumPy 1.24.3, SciPy 1.11.1
Key Libraries	Transformers 4.35.0
	TimER (timer-base-84m)

**Table 5 sensors-25-04208-t005:** Hyperparameter settings.

Component	Parameter	Value
Stage 1: TimerXL	Window size	32
	Detection rate	6–8%
	Min distance	4 steps
	Composite weights	[0.35, 0.45, 0.2]
Stage 2: DeBERTa	Hidden dimension	768
	Layers	8
	Attention heads	16
	Dropout	0.15
	Learning rate	$1\times 10^{-5}$
	Batch size	8
	Epochs	50
	Sequence length	32
Recovery Module	Prediction heads	2 (anomaly + recovery)
	Recovery loss weight	0.3
	TimER lookback	96 h

**Table 6 sensors-25-04208-t006:** Detailed performance metrics with statistical significance.

Method	Precision	Recall	F1-Score	Recovery MAE	*p*-Value *	Improvement
ARIMA	0.526 ± 0.208	0.682 ± 0.324	0.576 ± 0.161	6.375	<0.001	baseline
LSTM-AE	0.167 ± 0.073	0.097 ± 0.106	0.121 ± 0.077	0.126	<0.001	−79.0%
Anomaly Trans.	0.094 ± 0.042	0.106 ± 0.112	0.098 ± 0.066	0.356	<0.001	−83.0%
TimesNet	0.052 ± 0.015	0.267 ± 0.239	0.087 ± 0.099	0.164	<0.001	−84.9%
GNN	0.033 ± 0.000	0.300 ± 0.049	0.060 ± 0.001	N/A ^†^	<0.001	−89.6%
Lag-LLaMA	0.780 ± 0.089	0.823 ± 0.101	0.801 ± 0.087	0.2598	<0.001	+38.9%
EnhAT	0.951 ± 0.125	0.841 ± 0.181	0.873 ± 0.114	0.0055	-	+51.4%

* Paired *t*-test comparing to EnhAT, Bonferroni corrected. ${\,}^{†}$ Recovery MAE not calculated for GNN due to unstable predictions preventing reliable recovery estimation.

**Table 7 sensors-25-04208-t007:** Stagewise performance breakdown.

Configuration	Precision	Recall	F1-Score	Accuracy	AUC-ROC	Recovery MAE
Stage 1 Only	0.087 ± 0.015	0.793 ± 0.112	0.155 ± 0.023	0.524 ± 0.045	0.607 ± 0.032	N/A
Stage 2 Only	0.587 ± 0.089	0.961 ± 0.045	0.720 ± 0.067	0.775 ± 0.056	0.774 ± 0.041	0.0187
Complete System	0.951 ± 0.125	0.841 ± 0.181	0.873 ± 0.114	0.912 ± 0.078	0.925 ± 0.045	0.0055
Verification Rate	54.8% of Stage 1 detections verified by Stage 2					
False Positive Reduction	73.4% reduction from Stage 1 to final output					
Recovery Improvement	70.6% improvement in MAE with two-stage integration					

**Table 8 sensors-25-04208-t008:** Detailed ablation study results.

Configuration	F1-Score	ΔF1	Precision	Recall	MCC	*p*-Value ${\,}^{†}$
Complete System	0.873 ± 0.114	-	0.951 ± 0.125	0.841 ± 0.181	0.863 ± 0.091	-
*w*/*o* Balanced Loss	0.767 ± 0.089	−12.1%	0.800 ± 0.112	0.750 ± 0.156	0.537 ± 0.078	<0.001
*w*/*o* Ensemble Voting	0.570 ± 0.067	−34.7%	0.590 ± 0.089	0.653 ± 0.134	0.224 ± 0.045	<0.001
*w*/*o* Frequency Features	0.648 ± 0.078	−25.8%	0.587 ± 0.101	0.961 ± 0.067	0.709 ± 0.067	<0.001
*w*/*o* Sample Weighting	0.739 ± 0.089	−15.3%	0.630 ± 0.134	0.910 ± 0.089	0.682 ± 0.089	<0.001
*w*/*o* Augmentation	0.756 ± 0.101	−13.4%	1.000 ± 0.000	0.617 ± 0.156	0.614 ± 0.101	<0.001
*w*/*o* TimER Integration	0.812 ± 0.095	−7.0%	0.901 ± 0.089	0.789 ± 0.123	0.798 ± 0.078	<0.001
*w*/*o* Pretrained DeBERTa	0.698 ± 0.112	−20.0%	0.712 ± 0.134	0.701 ± 0.156	0.589 ± 0.101	<0.001

${\,}^{†}$ Wilcoxon signed-rank test with Bonferroni correction (α = 0.01).

**Table 9 sensors-25-04208-t009:** Detailed computational performance metrics.

Component	Mean (ms)	Std (ms)	% Total	Complexity
Data Loading	2.1	0.3	3.2%	O(n)
Feature Extraction	12.3	2.1	18.5%	O(n·d·w)
Stage 1: TimerXL	8.7	1.5	13.1%	O(n·d)
Stage 2: DeBERTa	45.6	5.3	68.5%	O(n·L·d^2^)
Recovery Module	3.2	0.4	4.8%	O(n·d)
Total Pipeline	66.6	7.2	100%	-
Throughput	900 samples/min (15 samples/second)			
GPU Utilization	42% (RTX 4060)			
Memory Usage	2.3 GB peak			

**Table 10 sensors-25-04208-t010:** Multipoint tampering detection performance with false merge analysis.

Tampering Pattern	Detection Rate	Precision	F1-Score	False Merge Rate
Single point	100%	0.951 ± 0.125	0.873 ± 0.114	N/A ^a^
2–3 consecutive	96%	0.923 ± 0.089	0.856 ± 0.078	8%
4–6 consecutive	92%	0.895 ± 0.101	0.831 ± 0.089	15%
Multiple non-consecutive	94%	0.912 ± 0.078	0.845 ± 0.067	3%
Mixed patterns	91%	0.887 ± 0.089	0.823 ± 0.078	11%

^a^ N/A: Not applicable for single-point anomalies as merging cannot occur with only one detection point.

**Table 11 sensors-25-04208-t011:** Detailed comparison with state-of-the-art methods.

Method	Year	F1-Score	Precision	Recall	Recovery MAE	Inference (ms)	Memory (GB)
ARIMA [5]	Traditional	0.576	0.526	0.682	6.375	3.2	0.1
LSTM-AE [12]	2016	0.121	0.167	0.097	0.126	12.5	0.5
GNN-based [23]	2020	0.060	0.033	0.300	N/A	23.7	1.2
Anomaly Transformer [16]	2021	0.098	0.094	0.106	0.356	78.9	2.1
TimesNet [17]	2022	0.087	0.052	0.267	0.164	92.3	2.5
Lag-LLaMA [18]	2024	0.801	0.780	0.823	0.2598	85.7	2.8
EnhAT (this work)	2025	0.873	0.951	0.841	0.0055	66.6	2.3

## Data Availability

The experimental code and preprocessed datasets are available at https://open-power-system-data.org/. The Open Power System Data (OPSD) Household Data package (version 2020-04-15) used for recovery experiments is publicly available under Creative Commons Attribution International license.

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
