# Peer review of "Optimized Two-Stage Anomaly Detection and Recovery in Smart Grid Data Using Enhanced DeBERTa-v3 Verification System"

_sensors, 2025, doi:10.3390/s25134208_

Round 1
Reviewer 1 Report
Comments and Suggestions for Authors
This study presents a novel hybrid Transformer architecture (Enhanced Anomaly Transformer) for anomaly detection and recovery in power grid data. The paper is well written and structured. However, there are still some scopes identified which will enhance the quality of the work further.
1. In the abstract, present outcome of the study using numerical data.
2. In introduction and related work it is necessary to highlight the gaps of the existing studies aligned with the contributions of the study.
3. In related work section add a table to summarize the benefits and drawbacks of the earlier study to strengthen the backbone of the literature.
4. In Section 3, for all the algorithms draw separate flowchart for better understanding.
5. The data source information is not well presented.
6. The results are not sufficient for validating the performance of the proposed algorithms.
7. The authors should compare their obtained results with the literature they mentioned in related work.
8. Sensitivity analysis should be added.
9. Avoid use of we/us/our.
10. Conclusions should highlight the major findings of the study aligning with the contributions.
Author Response
Comments 1: In the abstract, present outcome of the study using numerical data.
Response 1: Thank you for this valuable suggestion. We have revised the abstract to include comprehensive numerical results. The updated abstract now presents:
- Detection performance: F1-score of 0.873±0.114, representing 51.4% improvement over ARIMA
- Recovery accuracy: MAE of only 0.0055 kWh, representing 99.91% improvement
- Comparative results: 621% improvement over LSTM-AE, 791% over standard Anomaly Transformer
- Real-time capability: 66.6±7.2ms inference time
- Robustness metrics: detection accuracy above 88% across anomaly magnitudes
Location: Abstract, lines 12-25
Comments 2: In introduction and related work it is necessary to highlight the gaps of the existing studies aligned with the contributions of the study.
Response 2: We agree with this important observation. We have extensively revised both sections to clearly identify research gaps:
In Introduction (pages 2-3):
- Added explicit discussion of limitations: "Traditional anomaly detection methods suffer from fundamental limitations..."
- Identified specific gaps: precision-recall trade-off, lack of recovery capabilities, limited exploration of language models
- Aligned each contribution with identified gaps
In Related Work (Section 2, pages 4-8):
- Added concluding paragraph explicitly listing critical gaps
- Structured each subsection to highlight limitations of existing approaches
- Added direct connections between gaps and our contributions
Comments 3: In related work section add a table to summarize the benefits and drawbacks of the earlier study to strengthen the backbone of the literature.
Response 3: Thank you for this excellent suggestion. We have added Table 1 (page 5) providing a comprehensive comparison of anomaly detection methods across different categories:
- Statistical Methods (ARIMA, Exponential Smoothing, Kalman Filtering)
- Machine Learning (One-Class SVM, Isolation Forest, LOF, GNN-based)
- Deep Learning (LSTM-AE, CNN-based, GAN-based, VAE-based)
- Transformer-based (Anomaly Transformer, TimesNet, Our Method)
Each entry includes representative works, advantages, and limitations.
Comments 4: In Section 3, for all the algorithms draw separate flowchart for better understanding.
Response 4: We have added comprehensive flowcharts for all algorithms:
- Figure 4: TimerXL Multi-dimensional Anomaly Detection Flowchart (page 15)
- Figure 5: Comprehensive Feature Extraction Flowchart (page 17)
- Figure 6: Enhanced Anomaly Transformer Architecture Flowchart (page 18)
- Figure 7: Tampering Detection with TimER Flowchart (page 21)
- Figure 8: Data Recovery Process Flowchart (page 23)
Each flowchart uses standard notation with clear process flows, decision points, and annotations.
Comments 5: The data source information is not well presented.
Response 5: We have significantly expanded the dataset description in Section 4.1 with Table 3 (page 26):
- Total samples: 2,000 hourly measurements
- Training/test split: 80/20
- Features: Detailed descriptions of all three dimensions
- Date range: January 2020 - December 2021
- Source: Industrial AMI systems, Germany
- Load ranges: Specific kWh ranges for each feature
- Preprocessing steps: Quality validation, normalization, window extraction
Comments 6: The results are not sufficient for validating the performance of the proposed algorithms.
Response 6: We have substantially expanded the Results section (Section 5) with:
- Table 5: Detailed performance metrics with statistical significance (p-values)
- Table 6: Stage-wise performance breakdown
- Table 7: Comprehensive ablation study results
- Table 9: Multi-point tampering robustness analysis
- Figure 9: Multi-metric performance visualization
- Figure 10: Stage-wise analysis
- Figure 11: Training dynamics
- Statistical validation using paired t-tests with Bonferroni correction
Comments 7: The authors should compare their obtained results with the literature they mentioned in related work.
Response 7: We have added comprehensive comparisons:
- Table 5 (page 30): Direct comparison with ARIMA, LSTM-AE, Anomaly Transformer, TimesNet, GNN, and Lag-LLaMA
- Table 10 (page 38): Detailed comparison with state-of-the-art methods including year, performance metrics, and computational requirements
- All comparisons include statistical significance testing (p < 0.001)
- Explicit improvement percentages over each baseline method
Comments 8: Sensitivity analysis should be added.
Response 8: We have added comprehensive sensitivity analysis:
- Table 8 (page 37): Robustness analysis across different tampering scenarios
- Varying tampering magnitudes: 5-100 kWh
- Multiple tampering patterns: spike, drift, oscillation, step changes
- Position sensitivity: beginning, middle, end of sequence
- Figure 12: Visual representation of robustness across scenarios
- Recovery performance remains stable (relative error < 2.1%) across all conditions
Comments 9: Avoid use of we/us/our.
Response 9: We have revised the entire manuscript to use passive voice and third-person constructions throughout. Examples:
- "We propose" → "This study presents"
- "Our approach" → "The proposed approach"
- "We found" → "The results demonstrate"
All personal pronouns have been removed from the technical content.
Comments 10: Conclusions should highlight the major findings of the study aligning with the contributions.
Response 10: We have completely revised the Conclusions section to explicitly highlight:
- Key scientific contributions (5 main points)
- Quantitative achievements: F1-score 0.873, Recovery MAE 0.0055 kWh
- Practical implications: 95.1% precision, 84.1% recall
- Real-world applicability: 66.6ms inference time
- Direct alignment with the four main contributions stated in the introduction
- Future research directions based on findings
4. Additional clarifications
We have also made several additional improvements based on careful review:
- Added a Nomenclature table for all abbreviations and symbols
- Enhanced figure quality and clarity
- Improved mathematical notation consistency
- Added implementation details for reproducibility
- Included hardware specifications for performance context
We believe these revisions have significantly strengthened the manuscript and addressed all the reviewer's concerns comprehensively. Thank you again for your valuable feedback.

Reviewer 2 Report
Comments and Suggestions for Authors
The manuscript proposes the use of a hybrid transformer architecture for dealing with cyber-attacks on power grid information systems. (Enhanced Anomaly Transformer) for anomaly detection and recovery in power grid data. The model innovatively combines a pre-trained time series feature extractor with standard multi-head self-attention, overcoming limitations of traditional anomaly detection methods which often suffer from high false positive rates, poor adaptability to non-stationary time series data, and inability to effectively model contextual anomalies in complex multivariate settings. The pre-trained feature extractor, based on an improved DeBERTa-v3 architecture, effectively captures long-term dependencies in power data, while the multi-head self-attention with fixed parameters provides computational efficiency and consistent performance. The model was tested and results show significant improvements over the traditional ones.
The manuscript is interesting, well-written, and with valuable results. Still, some improvements are possible. Here are some comments and suggestions:
- There are many variables and abbreviations used in the manuscript. For easier reading, a Nomenclature list, placed in front of the text will be useful.
- In the Introduction section, several cases of cyber-attacks have been mentioned. However, no reference papers for these cases have been cited. Add some references for support of these statements.
- Add the reference paper for the Lag-LLaMA model.
- It is not clear how the training process has been organized. How the real measurement data were organized to fit the used models? Add some more explanations in the text.
- Eq.( 2): Define the value of d.
- Eqs. (2) and (3): Explain how the tempered point is distinguished from regular data. What represents the "significant anomalous increments in power consumption values"?
- Eq. (4): What is the meaning of "2" on the right part of the equation? Check it and correct it, if necessary.
- Explain how the anomaly is defined and detected. What are the criteria for the detection? Add some more explanation in the text.
Author Response
Response to Reviewer 2 Comments
1. Summary
Thank you very much for taking the time to review this manuscript. Please find the detailed responses below and the corresponding revisions/corrections highlighted/in track changes in the re-submitted files. Your insightful comments have helped us improve the clarity and technical rigor of our work significantly.
2. Questions for General Evaluation
|
Question |
Reviewer's Evaluation |
Response and Revisions |
|
Is the work a significant contribution to the field? |
Yes |
Thank you for recognizing the significance of our contribution. |
|
Is the work well organized and comprehensively described? |
Can be improved |
We have enhanced organization with a nomenclature list and improved clarity throughout. |
|
Is the work scientifically sound and not misleading? |
Yes |
We appreciate your assessment of the scientific validity. |
|
Are there appropriate and adequate references to related and previous work? |
Can be improved |
We have added all requested references, particularly for cyber-attacks and models. |
|
Is the English used correct and readable? |
Yes |
Thank you for your positive evaluation of the language quality. |
3. Point-by-point response to Comments and Suggestions for Authors
Comments 1: There are many variables and abbreviations used in the manuscript. For easier reading, a Nomenclature list, placed in front of the text will be useful.
Response 1: Thank you for this excellent suggestion. We have added a comprehensive Nomenclature table immediately after the abstract (page 3) that includes:
- All abbreviations (AMI, ARIMA, CNN, DeBERTa, etc.)
- Full descriptions for each term
- Organized alphabetically for easy reference
- Total of 18 key abbreviations defined
This significantly improves readability and serves as a quick reference for readers.
Comments 2: In the Introduction section, several cases of cyber-attacks have been mentioned. However, no reference papers for these cases have been cited. Add some references for support of these statements.
Response 2: We agree completely. We have added specific references for all cyber-attack cases mentioned:
- 2019 malware infections at India's nuclear power plants: Singh et al. (2020) [ref-cyber-1]
- Venezuela's electrical system attacks: Musleh et al. (2020) [ref-cyber-2]
- 2020 ransomware attacks on Brazilian power utilities: Mohammadi et al. (2021) [ref-cyber-3]
Location: Introduction, page 2, with full citations in the References section
Comments 3: Add the reference paper for the Lag-LLaMA model.
Response 3: Thank you for noting this omission. We have added the proper reference:
- Rasul et al. (2024) "Lag-LLaMA: Towards foundation models for time series forecasting" [ref-lagllama]
This reference appears in:
- Section 2.3 (page 6) when first introducing Lag-LLaMA
- Throughout the paper where Lag-LLaMA results are discussed
- Complete citation in References section
Comments 4: It is not clear how the training process has been organized. How the real measurement data were organized to fit the used models? Add some more explanations in the text.
Response 4: We have significantly expanded Section 4.2 "Training Process Organization" with a detailed five-phase approach:
Phase 1: Data Preparation
- Load 2,000 hourly measurements (24 months)
- Stratified 80/20 split preserving seasonal characteristics
- Baseline statistics computation
Phase 2: Stage 1 Detector Configuration
- Increment-based statistics calculation
- Threshold calibration (P90, P95, P99)
- Composite scoring weight optimization
Phase 3: Synthetic Anomaly Generation
- 100 training scenarios
- 3-6 anomalies per scenario
- Minimum 15-step separation
- Balanced anomaly type representation
Phase 4: Neural Network Training
- 32-step temporal window extraction
- 75-dimensional feature computation
- Balanced loss function with recovery component
- Data augmentation (20% probability)
Phase 5: System Integration
- End-to-end pipeline validation
- 15 independent test scenarios
- Performance measurement
Location: Section 4.2, pages 27-28
Comments 5: Eq.(2): Define the value of d.
Response 5: We have clarified that d=3 represents the three dimensions of power measurements:
- DE_KN_industrial1_grid_import
- DE_KN_industrial1_pv_1
- DE_KN_industrial3_compressor
This is now explicitly stated immediately after Equation (2) in Section 3.2.1.
Comments 6: Eqs. (2) and (3): Explain how the tempered point is distinguished from regular data. What represents the "significant anomalous increments in power consumption values"?
Response 6: We have added detailed explanations:
For distinguishing tampered points: The detection criterion for significant anomalous increments is formally defined as: |Δx_t| > μ_local + 2.5σ_local
where μ_local and σ_local are computed over a 48-hour sliding window.
Four anomaly types are defined:
- Spike anomalies: intensity ∈ [7.0, 15.0] × standard deviation
- Drift anomalies: gradual changes over 8-15 time steps
- Oscillation anomalies: periodic patterns with f ∈ [0.2, 0.5]
- Step changes: sudden level shifts with partial recovery
Location: Section 3.5.1 "Anomaly Types and Detection Criteria" (page 24)
Comments 7: Eq.(4): What is the meaning of "2" on the right part of the equation? Check it and correct it, if necessary.
Response 7: Thank you for catching this error. We have reviewed and corrected Equation (4). The equation now properly shows: S_extreme(t) = Σ w_i · |Δx_t|/(Threshold_i + ε)
The "2" was indeed an error and has been removed. The weights w = [0.4, 0.3, 0.3] and thresholds = [P_95, IQR, P_90] are now clearly defined.
Comments 8: Explain how the anomaly is defined and detected. What are the criteria for the detection? Add some more explanation in the text.
Response 8: We have added comprehensive explanations in multiple sections:
Detection Process (Section 3.3):
- Two-stage approach: high-recall initial detection + high-precision verification
- TimER-based verification with MSE threshold = 10.0
- Algorithm 3 provides step-by-step detection procedure
Detection Criteria:
- Stage 1: Composite score combining extreme value, peak, and consistency detection
- Stage 2: Ensemble voting with multiple thresholds
- Verified if: (1) ≥2 votes from threshold ensemble, or (2) p > 0.7 AND confidence > 0.2
Anomaly Definition:
- Formal definition: Points where observed increments significantly deviate from expected patterns
- Quantitative threshold: MSE > 10.0 between predicted and actual values
- Physical constraints: non-negative consumption, operational range limits
Location: Sections 3.3, 3.4, and 3.5
4. Additional clarifications
We have also made several improvements to enhance clarity:
- Added detailed captions to all figures explaining what they show
- Included specific numerical values in all result discussions
- Provided implementation details for reproducibility
- Enhanced mathematical notation consistency throughout
Thank you again for your thorough review and constructive feedback. These changes have substantially improved the technical clarity and presentation of our work.

Reviewer 3 Report
Comments and Suggestions for Authors
The paper proposes a novel hybrid Transformer architecture for anomaly detection and recovery in power grid data. For proving its efficiency experiments and analysis based on real data-sets is proposed. Overall the paper has an apropriate structure and content. Despite of these remarks some editing and content issues, few listed below are recommended to be addressed.
Along mentioned points it is recommended to the authors to analyze if it is proper to refer to "power grid" when experiments are related to data from "electric distribution grid".
Described tests and discussions are performed on one point tampered data. Is it realistic? How proposed algorithm perform if more consecutive data or not are tampered?
Line 1: In "The reliability and integrity of power data have become critical factors ..." check if more clearer term could be used instead of "power data";
lines 12-13: In " ... detects tampered data points with a clear anomaly signal (MSE of 84.48) and achieves remarkably precise recovery with a mean absolute error (MAE) of ..." it is recommended to maintain consistency in presentation approach, eg. MAE is explained while Mean Squared Error not ... Also reported result of "only 0.0055 kWh" is impressive but could be misleading if amplitude is close to this value. So if possible other approach to be used is recommended (eg. relative units/percentage, etc.);
Lines 15, 203, etc.: idem. in case of "... achieved an MAE of 0.2598 kWh ...";
Lines 33-46: it is recommended to suport this part with relevant references;
Lines 54-56: it is recommended to suport this part with relevant references or mentioning a methodology were determined;
Line 123: at their first occurrence acronyms/abbreviations are recommended to be defined (see TFT. Check if this note applies in case of the few lines above; CNN, LSTM, GAN, etc. structures or later eg line 275 for TS-TCC);
Lines 170-171: please check the observation related to Abstract (lines 12-13). Also check if it is relevant to specify details already mentioned.
In Fig. 1 Label from left block, "Overall Architecture", is recommended to be revised to have a more suggestive term; check if possible to avoid elements of figure to overlap (see arrows and label "Disentangled Attention Computation");
In both Fig. 1 and 2 numerical value of MAE is imposed to architecture or is experimental obtained value? Should be that mentioned?
Lines 210-211; In "More recent deep learning approaches for data recovery include GAN-based imputation [35] and recurrent neural networks with attention [36]." for audience familiarized with ML would be fine, but could be confusing for other readers, so a slightly reformulation could be considered, eg. GAN-based imputing missing data;
line 257: check for proper capitalization "indicates tampering, and For identified tampered";
lines 321-322: In "In our implementation, we use a lookback length l of 96 time points (corresponding to 96 hours in hourly resolution data) ..." a bit could be confusing for a reader. Author meant about data corresponding to 24 h with a resolution of 4 points per hour (each at 15 min.)?
line 397: idea already mentioned few times previously. It is recommended to be checked harmonized to avoid repetition.
In Fig. 5 non-Latin characters are present (see labels on each abscisa);
Author Response
Thank you very much for taking the time to review this manuscript. Please find the detailed responses below and the corresponding revisions/corrections highlighted/in track changes in the re-submitted files. Your detailed editorial comments have helped us significantly improve the presentation quality and clarity of our manuscript.
2. Questions for General Evaluation
|
Question |
Reviewer's Evaluation |
Response and Revisions |
|
Is the work a significant contribution to the field? |
Yes |
Thank you for recognizing the significance of our contribution. |
|
Is the work well organized and comprehensively described? |
Can be improved |
We have addressed all organizational and presentation issues as detailed below. |
|
Is the work scientifically sound and not misleading? |
Yes |
We appreciate your positive assessment of the scientific soundness. |
|
Are there appropriate and adequate references to related and previous work? |
Can be improved |
We have added all requested references and support statements. |
|
Is the English used correct and readable? |
Can be improved |
We have corrected all language and formatting issues identified. |
3. Point-by-point response to Comments and Suggestions for Authors
Comments 1: It is recommended to the authors to analyze if it is proper to refer to "power grid" when experiments are related to data from "electric distribution grid".
Response 1: Thank you for this important distinction. After careful consideration, we have maintained "power grid" as our terminology because:
- Our dataset includes both consumption and generation data (including photovoltaic generation)
- The methods are applicable to the broader power grid infrastructure, not limited to distribution
- The cyber-security implications affect the entire grid system
- Industry literature commonly uses "smart grid" to encompass both transmission and distribution
However, we have added clarification in Section 4.1 that our experimental data specifically comes from distribution-level measurements within the broader power grid context.
Comments 2: Described tests and discussions are performed on one point tampered data. Is it realistic? How proposed algorithm perform if more consecutive data or not are tampered?
Response 2: Excellent point. We have added comprehensive multi-point tampering analysis:
New Section 5.6: Multi-point Tampering Robustness Analysis
- Table 9 shows performance across different tampering patterns:
- Single point: 100% detection, 0.0055 MAE
- 2-3 consecutive: 96% detection, 0.0089 MAE
- 4-6 consecutive: 92% detection, 0.0123 MAE
- Multiple non-consecutive: 94% detection, 0.0098 MAE
- Mixed patterns: 91% detection, 0.0145 MAE
The analysis confirms robust performance even with sophisticated multi-point attacks, with graceful degradation for longer consecutive tampering.
Location: Section 5.6, pages 36-37
Comments 3: Line 1: In "The reliability and integrity of power data have become critical factors ..." check if more clearer term could be used instead of "power data"
Response 3: We have revised this to be more specific: "The reliability and integrity of smart grid measurements have become critical factors..."
This clarifies that we're referring to measurement data from smart grid infrastructure rather than the vague term "power data."
Comments 4: Lines 12-13: maintain consistency in presentation approach, eg. MAE is explained while Mean Squared Error not
Response 4: We have revised for consistency. The abstract now:
- Defines both metrics at first use: "Mean Squared Error (MSE)" and "Mean Absolute Error (MAE)"
- Uses consistent format throughout: "MSE of 84.48" and "MAE of 0.0055 kWh"
- Maintains this consistency throughout the entire manuscript
Comments 5: Lines 15, 203, etc.: idem. in case of "... achieved an MAE of 0.2598 kWh ..."
Response 5: We have ensured consistency by:
- Always including units: "MAE of 0.2598 kWh" (not just "0.2598")
- Using the same format for all metrics
- Defining abbreviations at first use in each major section
- Maintaining consistent precision (4 decimal places for MAE values)
Comments 6: Lines 33-46: it is recommended to support this part with relevant references
Response 6: We have added supporting references for all claims:
- Industry attack statistics: Liang et al. (2017) [ref-journal-1]
- Traditional method limitations: Box et al. (2015) [ref-journal-9], Hyndman & Athanasopoulos (2018) [ref-journal-10]
- Deep learning advances: Vaswani et al. (2017) [ref-journal-5], Zhou et al. (2021) [ref-journal-6]
Comments 7: Lines 54-56: it is recommended to support this part with relevant references or mentioning a methodology were determined
Response 7: We have added:
- Reference to methodology: "determined through analysis of industry incident reports [ref-journal-1]"
- Supporting statistics from cited sources
- Clear methodology statement: "These statistics were compiled from publicly reported incidents between 2019-2024"
Comments 8: Line 123: at their first occurrence acronyms/abbreviations are recommended to be defined
Response 8: We have systematically reviewed and corrected all acronym definitions:
- TFT: "Temporal Fusion Transformer (TFT)"
- CNN: "Convolutional Neural Network (CNN)"
- LSTM: "Long Short-Term Memory (LSTM)"
- GAN: "Generative Adversarial Network (GAN)"
- TS-TCC: "Time Series Temporal and Contextual Contrasting (TS-TCC)"
All are now defined at first occurrence with the acronym in parentheses.
Comments 9: Lines 170-171: please check the observation related to Abstract (lines 12-13). Also check if it is relevant to specify details already mentioned.
Response 9: We have removed redundant information and ensured that:
- Numerical results are presented consistently
- Details are not unnecessarily repeated
- Each section provides new information or context
- Cross-references are used instead of repetition
Comments 10: In Fig. 1 Label from left block, "Overall Architecture", is recommended to be revised to have a more suggestive term
Response 10: We have revised the figure labels:
- "Overall Architecture" → "Two-Stage Detection Framework"
- Added more descriptive labels throughout
- Ensured no overlapping elements (arrows and labels are clearly separated)
Comments 11: In both Fig. 1 and 2 numerical value of MAE is imposed to architecture or is experimental obtained value? Should be that mentioned?
Response 11: We have clarified in the figure captions:
- Figure 1: "MAE values shown are experimentally obtained results from our evaluation"
- Figure 2: "Performance metrics represent actual experimental measurements, not architectural parameters"
- Added notes distinguishing between architectural components and performance results
Comments 12: Lines 210-211: could be confusing for other readers, so a slightly reformulation could be considered
Response 12: We have reformulated for clarity: Original: "GAN-based imputation and recurrent neural networks with attention" Revised: "GAN-based methods for imputing missing data and recurrent neural networks enhanced with attention mechanisms"
This clarifies the purpose and makes it accessible to readers less familiar with ML terminology.
Comments 13: Line 257: check for proper capitalization "indicates tampering, and For identified tampered"
Response 13: Corrected to: "indicates tampering, and for identified tampered points..." We have also conducted a comprehensive review of capitalization throughout the manuscript.
Comments 14: Lines 321-322: In "In our implementation, we use a lookback length l of 96 time points (corresponding to 96 hours in hourly resolution data) ..." a bit could be confusing for a reader.
Response 14: We have clarified: "In our implementation, we use a lookback length l of 96 time points. Since our data has hourly resolution (one measurement per hour), this corresponds to 96 hours or 4 days of historical data."
This makes the time resolution relationship explicit and clear.
Comments 15: Line 397: idea already mentioned few times previously. It is recommended to be checked harmonized to avoid repetition.
Response 15: We have removed repetitive statements and consolidated key points to appear only once in appropriate sections. The manuscript now presents ideas progressively without unnecessary repetition.
Comments 16: In Fig. 5 non-Latin characters are present (see labels on each abscissa)
Response 16: We have regenerated all figures to ensure:
- Only Latin characters are used
- All labels are clearly readable
- Consistent font usage throughout
- Proper encoding for all text elements
4. Additional clarifications
We have also implemented several general improvements:
- Conducted a complete editorial review for consistency
- Verified all figure quality and text rendering
- Ensured proper formatting of all equations
- Double-checked all numerical values for consistency
- Added page numbers to all table/figure references
Thank you for your meticulous review. These editorial improvements have significantly enhanced the professional presentation of our work.

Round 2
Reviewer 1 Report
Comments and Suggestions for Authors
No more comments.